# Sen2Like: Paving the Way towards Harmonization and Fusion of Optical Data

Sébastien Saunier [1,*], Bringfried Pflug [2], Italo Moletto Lobos [3], Belen Franch [3,4], Jérôme Louis [1], Raquel De Los Reyes [2], Vincent Debaecker [1], Enrico G. Cadau [5], Valentina Boccia [6], Ferran Gascon [6] and Sultan Kocaman [7,8]

1   Telespazio France, Satellite System and Operation, 26 Avenue JF Champollion, BP 52309, CEDEX 1, 31023 Toulouse, France
2   German Aerospace Center (DLR), Remote Sensing Technology Institute, Photogrammetry and Image Analysis, Oberpfaffenhofen, 82234 Wessling, Germany
3   Global Change Unit, Image Processing Laboratory, University of Valencia, Paterna, 46980 Valencia, Spain
4   Department of Geographical Sciences, University of Maryland, College Park, MD 20742, USA
5   Serco Italia S.p.A—Via Sciadonna 24–26, 00044 Frascati, Italy
6   European Space Agency, Directorate of Earth Observation Programmes, Largo Galileo Galilei 1, 00044 Roma, Italy
7   Department of Geomatics Engineering, Hacettepe University, Beytepe, Ankara 06800, Turkey
8   Photogrammetry and Remote Sensing, ETH Zurich, 8093 Zurich, Switzerland
*   Correspondence: sebastien.saunier@telespazio.com; Tel.: +33-534-357-501

**Abstract:** Satellite Earth Observation (EO) sensors are becoming a vital source of information for land surface monitoring. The concept of the Virtual Constellation (VC) is gaining interest within the science community owing to the increasing number of satellites/sensors in operation with similar characteristics. The establishment of a VC out of individual missions offers new possibilities for many application domains, in particular in the fields of land surface monitoring and change detection. In this context, this paper describes the Copernicus Sen2Like algorithms and software, a solution for harmonizing and fusing Landsat 8/Landsat 9 data with Sentinel-2 data. Developed under the European Union Copernicus Program, the Sen2Like software processes a large collection of Level 1/Level 2A products and generates high quality Level 2 Analysis Ready Data (ARD) as part of harmonized (Level 2H) and/or fused (Level 2F) products providing high temporal resolutions. For this purpose, we have re-used and developed a broad spectrum of data processing and analysis methodologies, including geometric and spectral co-registration, atmospheric and Bi-Directional Reflectance Distribution Function (BRDF) corrections and upscaling to 10 m for relevant Landsat bands. The Sen2Like software and the algorithms have been developed within a VC establishment framework, and the tool can conveniently be used to compare processing algorithms in combinations. It also has the potential to integrate new missions from spaceborne and airborne platforms including unmanned aerial vehicles. The validation activities show that the proposed approach improves the temporal consistency of the multi temporal data stack, and output products are interoperable with the subsequent thematic analysis processes.

**Keywords:** Sentinel-2; Landsat; surface reflectance; Virtual Constellation; Analysis Ready Data; harmonization; data fusion; Copernicus

## 1. Introduction

The Sentinel-2 (S2) mission managed by the European Space Agency (ESA) has been fully operational since June 2017 with a constellation of two polar orbiting satellite units. Both the Sentinel-2A (S2A) and Sentinel-2B (S2B) satellites are equipped with an optical imaging sensor, namely the Multi-Spectral Instrument (MSI), which acquires high spatial resolution images with ground sampling distances (GSDs) of 10 to 60 m depending on the

wavelength [1]. The S2 constellation is dedicated to land monitoring, emergency response and management and security applications. It is used for the monitoring of land cover change and biophysical variables related to agriculture and forestry, monitors coastal and inland waters and is useful for disaster risk assessment and management [2]. The S2 mission offers a 5 day revisit frequency, with the same viewing geometry, of any location on Earth.

Currently, the understanding and characterization of time-varying land changes at fine spatial scale is becoming a real concern. The challenge is to maximize revisit frequency irrespective of the Earth location, such as over equatorial and high latitude regions, and thus reduce the effect of limiting factors such as persistent cloud coverage. The combination of the Sentinel-2A/2B (S2A/B) and Landsat-8/9 (LS8/LS9) sensors together already provides a global median average revisit of 2.3 days [3]. Furthermore, since the 1970s, the average number of satellites launched per year/per decade has increased from 2 to 12, and the spatial resolution has increased from around 80 m to better than 1 m multispectral (MS) and better than 0.5 m for panchromatic bands [4], providing the opportunity to decrease the global median average revisit time.

The notion of a multi-mission spatiotemporal dataset is now widely democratized in the Earth Observation (EO) community. Data providers have been delivering dedicated products, called Analysis Ready Data (ARD, [5]), following the Committee on Earth Observation Satellites (CEOS) Analysis Ready Data for Land (CARD4L) guidance for some of them [6]. These products are ready to be employed within EO Data Cubes (EODC) implementations such as Digital Earth Australia, the Swiss Data Cube, the EarthServer, the E-sensing platform or the Google Earth Engine [7].

The handling of multi-mission MS data has several challenges due to the differences between sensor characteristics, product definitions and observation geometries that have a direct impact on the consistency and accuracy of spatiotemporal datasets. To overcome these issues, it is common to use cosmetic methods (post-processing) in order to improve the temporal data consistency and reduce the noise. This kind of approach has major drawbacks as it increases the uncertainties and might finally prevent the detection or full characterization of observed features, in particular in the context of land surface change analysis.

An alternative approach discussed herein is to apply processing steps dedicated to data harmonization (equivalent to radiometry) and data fusion (equivalent to pixel spacing) based on physical parameters and calibration data. Several academic and industrial actors such as NASA-HLS, FORCE, CESBIO-MAJA and the Planet Fusion Product [8–11] have been promoting this approach. In addition, they acknowledge that the geometric, atmospheric, spectral and directional effect corrections are fundamental processing stages to be performed prior to the thematic exploitation of multi-satellite information.

The main aim of this study is to propose a new algorithm for the harmonization and fusion of Landsat 8/9 and S2 MS products to increase the temporal resolution and provide a solution for continuous Earth surface monitoring. A software tool, namely Sen2Like (S2L), has been developed to implement the algorithm and to aid the VC establishment activities within EU Copernicus Program and funded by ESA/EU. The baseline principle of Sen2Like ([12]) was to consider S2 as a reference mission in terms of the instrument, product accuracy specification ([13]) and product format and to combine the S2 data stream with any other data sources delivered by MS missions with spectral characteristics similar to S2 sensors. The final objective is thus to create a Virtual Constellation (VC) as previously discussed by Wulder et al. [14]. The S2L is open source and developed in the Python programming language.

The S2L methodology analyzes the user products that are stored locally or available at the Copernicus Data and Information Access Service (DIAS) cloud environment and generates the output ARD dataset based on various user configurations such as temporal period, geographical footprint and cloud cover. The final S2L ARD format complies with the CARD4L guidelines [6] in terms of general and per pixel metadata information.

The S2L implementation involves a data harmonization protocol, and for this purpose, a processing framework is proposed. For any input mission, the processing sequence remains unchanged, including various methods dedicated to geometric and atmospheric issues, normalization of reflectance values, spectral band pass correction and downscaling.

The data harmonization and data fusion processing method proposed here is novel and modifies the original physical Top of Atmosphere (TOA) values substantially. Even if a full traceability chain does not exist in the current implementation, per pixel quality assurance information is provided. This article presents the S2L processing algorithms and describes the software characteristics. The ongoing data validation activities are briefly introduced, and the results are discussed as well. The S2L methods are adaptable to new sensors and platforms when calibration data are available, as addressed in Section 4. The final section of the article provides the conclusions of the study and major items of future work.

## 2. Materials and Methods

In this section, the processing methods and the S2L software design drivers are explained in detail. The main processing steps of the S2L algorithm are presented in Figure 1. The S2L software is plugged into a data archive (local or DIAS) and, depending on the user configuration, a catalog access module queries the expected set of Level-1 (L1) and Level-2 (L2) product samples. The set of S2L processes are then triggered over this sample. The L2H or L2F products are generated as a result of the spectral adjustment and data fusion. The processing workflows contain the following configurable steps depending on the input product type (i.e., S2 L1C, S2 L2A, S2 L2A MAIA, LS8/LS9 Collection 1/Collection 2), as can be seen in Figure 1:

i.      Image stitching of the different tiles;
ii.     Geometric corrections including the co-registration to a reference image;
iii.    Inter-calibration (for S2 L1C);
iv.     Atmospheric corrections;
v.      Transformation to Nadir Bidirectional Reflectance Distribution Function (BRDF) Adjusted Reflectance (NBAR);
vi.     Application of Spectral Band Adjustment Factor (SBAF) (for LS8/9);
vii.    Production of LS8/LS9 high resolution 10 m pixel spacing data (data fusion).

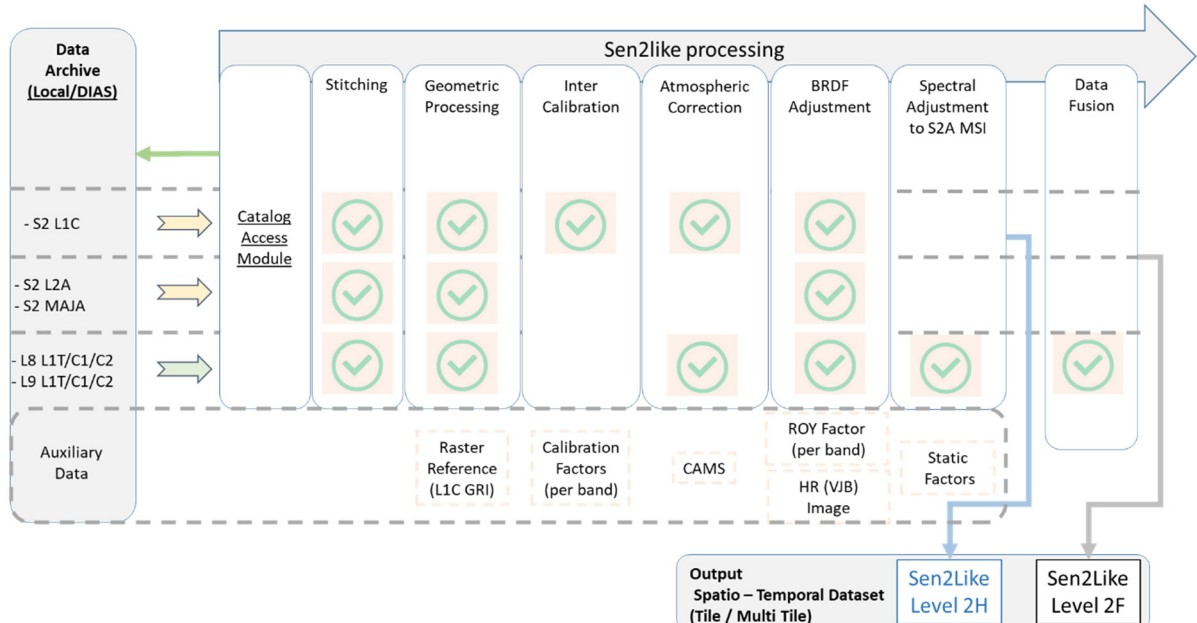

**Figure 1.** The Sen2Like processing framework for generating L2H and L2F products.

As a result, an L2 Harmonized (L2H) or L2 Fused (L2F) ARD dataset matching the Region of Interest (ROI) selected is generated. Regarding tiling, the S2L data are framed in tiles with a size of 110 km × 110 km in the Universal Transverse Mercator (UTM)-based Military Grid Reference System (MGRS, [15]). If the ROI extent covers several MGRS tiles, all of them are processed.

### 2.1. Sen2Like Software Design Elements

The S2L software is designed for on demand processing purposes with major requirements set on near real time production and the provision of harmonized data in a timely manner. In addition, any existing S2L spatiotemporal dataset can be regularly and quickly updated along with the new acquisition. Furthermore, the design enables researchers to configure the software depending on the accuracy and timeliness requirements and the processing system specification. By using a minimal configuration (e.g., local computer, basic processing), the generation of a single S2L product takes approximately 4–5 min.

The S2L framework was developed in the Python 3.7 programming environment, and the implementation utilizes recent data science libraries such as Numpy, Pandas, Xarray, Scipy, Scikit-Learn, etc. The framework architecture is modular and allows a fast integration of processing modules. From an operational perspective, the S2L architecture is process-oriented; i.e., for a single processing step, several methods exist and are optionally proposed to the user in a configurable way. This architecture has various advantages. From a functional point of view, it is the best tradeoff to facilitate the integration of new missions such as MS and hyperspectral missions as well as aerial cameras including those aboard unmanned aerial vehicles (UAVs). From a scientific point of view, it enables users to perform different accuracy assessments with the goal of methodological inter-comparisons. Finally, and more practically, the tool is tailored to run in a cloud environment and optimized for performance purposes such as per band parallelization functionality. On the other hand, when configured with simple methods, the tool can run standalone on a personal computer with low computational resources and low energy consumption.

### 2.2. Product Description and Auxiliary Data

The Sen2Like software ingests a stack of L1/L2 S2/LS8 and LS9 products (available from the Copernicus Access Hub [16], ESA LS8 web portal [17]) and provides two categories of the ARD dataset: i.e., the L2F and the L2H dataset. Both L2H and L2F products are processed in Standard Archive Format for Europe (SAFE) [18] and embed Cloud Optimized (COG) GeoTIFF images [19]. Furthermore, product metadata are also described with a dedicated Spatio-Temporal Asset Catalogs (STAC, [20]) JSON file to expose data.

The major difference between the two L2H and L2F product types resides in the spatial resolution of the output images; in the L2H products, the native spatial resolutions of input images are preserved, whilst in the L2F products, the resolutions of the LS8/LS9 image data are upsampled to the pixel spacing of the relevant S2 band. Table 1 depicts the harmonization and fusion approaches by presenting the relationship between the S2L band, the input sensor dependent band name and the pixel size of related L2F/L2H images. As shown in the table, depending on the S2L band, the respective S2 LS8/LS9 band central wavelength might be slightly different. S2L LS8/LS9 bands for which image fusion is applied are indicated in bold. The S2L bands that are not processed and therefore kept as "native" are indicated in italic.

**Table 1.** Existing image data embedded within the L2H and L2F Sen2Like SAFE products and spatial resolution.

| S2L Band | Band Designation | S2 MSI Bands (Center Wavelength [μm]) | LS8/LS9 OLI/TIRS Bands (Center Wavelength [μm]) | L2H S2 Resolu-tion(m) | L2H-LS8/LS9 Resolution (m) | L2F S2 Resolution (m) | L2F LS8/LS9 Resolution (m) |
|---|---|---|---|---|---|---|---|
| B01 | Coastal Aerosol | B01 (443 nm) | B01 (442 nm) | 60 | 30 | 60 | 30 |
| B02 | Blue | B02 (490 nm) | B02 (482 nm) | 10 | 30 | 10 | 10 |
| B03 | Green | B03 (560nm) | B03 (561 nm) | 10 | 30 | 10 | 10 |
| B04 | Red | B04 (665 nm) | B04 (654 nm) | 10 | 30 | 10 | 10 |
| B08 | NIR 1 | B08 (842 nm) | None [1] | 10 | - | 10 | - |
| B8A | NIR2 | B8A (865 nm) | B05 (864 nm) | 20 | 30 | 20 | 20 |
| B11 | SWIR 1 | B11 (1610 nm) | B06 (1608 nm) | 20 | 30 | 20 | 20 |
| B12 | SWIR 2 | B12 (2190 nm) | B07 (2200 nm) | 20 | 30 | 20 | 20 |
| BP1 | Panchromatic | None [2] | B08 (589 nm) | - | 15 | - | 15 |
| BT1 | TIRS 1 | None [2] | B10 (11 μm) | - | 100 | - | 100 |
| BT2 | TIRS 2 | None [2] | B11 (12.2 μm) | - | 100 | - | 100 |
| B05 | Red Edge 1 | B05 (705 nm) | None [1] | 20 | - | 20 | |
| B06 | Red Edge 2 | B06 (740 nm) | None [1] | 20 | - | 20 | |
| B07 | Red Edge 3 | B07 (783 nm) | None [1] | 20 | - | 20 | |

[1] No corresponding band in Landsat 8/Landsat 9 OLI data. [2] No corresponding band in Sentinel-2 MSI data.

The S2L processing uses a set of auxiliary data, with the respective quality influencing the quality of the final products. Among auxiliary data, the following issues are essential:

- The geometric reference data involved in the co-registration step;
- The Copernicus Atmosphere Monitoring Service (CAMS) data [21], meteorological data released by the European Centre for Medium-Range Weather Forecasts (ECMWF);
- The spectral adjustment parameter set (SBAF corrections);
- The calibration processing coefficient;
- The directional effects correction factors (NBAR corrections).

Further details on the input/output product formats and descriptions are provided in Appendix A.

## 2.3. Image Correction Methodology

In this section, the image processing methods including the geometric and atmospheric correction, the BRDF correction, SBAF and the data fusion approach are explained in detail.

### 2.3.1. Geometric Correction

The proposed S2L geometric correction methodology is basically categorized into four main components, which are (i) image stitching (mosaicking), (ii) MGRS reframing, (iii) geometric co-registration and (iv) quality control. The stitching and the MGRS reframing steps are essential to ensure full coverage of MGRS frame depending on L1/L2 product footprint conventions. A second goal of the S2L is to co-register all input images with respect to a raster reference. In this way, the geometrical consistency of the spatiotemporal dataset reaches an accuracy goal of 0.3 pixel (10 m) at 2-sigma depending on the radiometric quality and similarity of the data. The proposed S2L methodology for co-registration relies on dense image matching between search and reference images using the KLT (Kanade Lucas-Tomasi) algorithm [22,23] implemented in the OpenCV [24,25] library in Python. The feature points extracted with the Good Features algorithm of Shi and Tomasi [26] are employed in the process to increase the matching success and the reliability. This methodology was utilized in earlier projects [27–32] demonstrating that computer vision technics are beneficial for EO data processing. As shown in Figure 2, the geometric assessment results are obtained from the matching between near infrared (NIR) band images of both sensors and then extrapolated to all other bands for correction purposes.

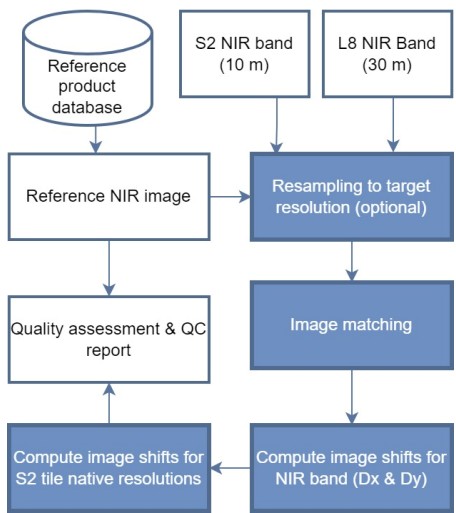

**Figure 2.** Geometric processing workflow including co-registration and geometric quality assessment.

Depending on the radiometric quality and similarity, approximately 100 ground control points (GCPs) disseminated over one MGRS scene footprint provide consistent statistical results. As discussed previously and also illustrated in Appendix A, the S2L ingests data from different processing centers and delivers products in their own format. Thus, the processing and accuracy specifications depend on the respective product level, software version and relative configuration.

### 2.3.2. Atmospheric Correction

The objective of atmospheric correction (AC) is to remove the scattering and absorption effects from the atmosphere in order to obtain a good estimate of the surface reflectance (SR). The AC might be time consuming and computationally demanding, but for some EO application use cases, a simplified modelling in support of AC fits the purpose. For this reason, two distinct AC approaches are proposed in the S2L.

The first AC methodology is based on a Simplified Model for the AC (SMAC) of satellite measurements in the solar spectrum [33]. It utilizes simple analytical formulas (based on the 5S model) with a set of 49 coefficients, which were fitted using a large number of radiative transfer simulations with the 6S model (not 6SV). The SMAC coefficients (CESBIO, [34]) were computed for each band of every instrument (i.e., OLI, MSI-A and MSI-B).

The parameters in the observation period, such as the atmospheric sea level pressure, the total aerosol optical depth at 550 nm, the ozone content and the water vapor content data, are obtained from the Atmosphere Data Store (ADS) maintained by CAMS [21], which are provided with a resolution of approximately 40 km. The meteorological conditions are assumed to be stable over the geographical extent of an S2 MGRS tile footprints. In addition, the atmospheric parameter set matching the MGRS tile scene center and product observation date/time is computed. Furthermore, the solar and viewing angle values are handled differently depending on their availability in the L1/L2 product format. A fall-back option consists of using FMASK [35] routines to calculate them. Obtained from their original grids, the viewing and solar viewing angles are upsampled to a finer grid, depending on the resolution of target image grid. For S2, the original grid is a $23 \times 23$ (5 km resolution), and for Landsat, the grid is finer with a spatial resolution of 300 m $\times$ 300 m.

As shown in Figure 3, the processing workflow uses the SMAC Look Up Table (LUT) involved in the estimation of a parametric model, subsequently used for TOA to SR corrections. The TOA to SR second-order polynomial function coefficients are estimated based on the SMAC unitary results. The SMAC is triggered on any TOA reflectance value within an interval from 0.01 to 1.0. The method does not account for terrain slope and

adjacency effects. For similar Landsat-5 TM bands 1 and 4, the maximum relative errors relative to original 5S code were estimated to be 1.65% and 2.37%, respectively [33]. The limits of SMAC are the same as for 5S/6S, i.e., solar and viewing angles must be less than 60 and 50 degrees, respectively, and the aerosol optical depth must be less than 0.8 at 550 nm [33].

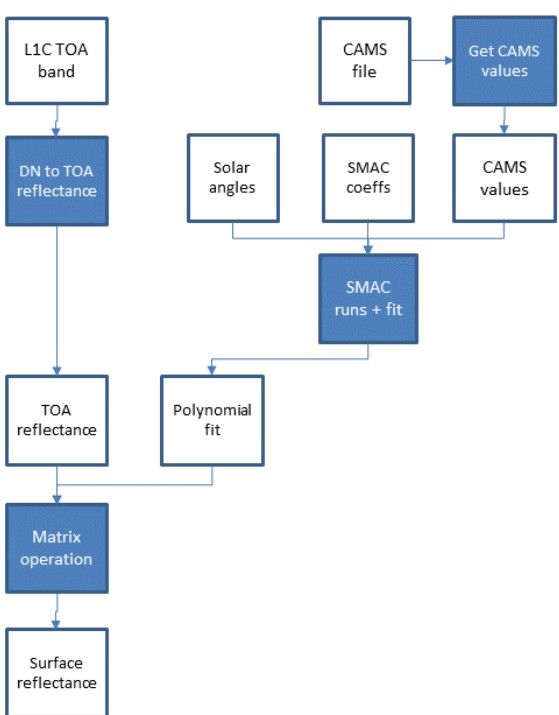

**Figure 3.** Sen2Like Fast Atmospheric Processing Workflow diagram by using SMAC LUT.

The second option offered by the S2L framework for AC is the use of the Sen2Cor 3.0 tool [36]. The Sen2Cor tool's performance is regularly reported in L2A Data Quality Reports [13] as well as in independent inter-comparison exercises such as the ACIX [37] and the ACIX-II. The average accuracy value relative to the average surface reflectance reference is below or near to 5% except for bands 5 and 12 [36].The Sen2Cor methods have a domain validity range where the solar zenith angle should be less than 70 degrees.

2.3.3. BRDF Correction

Two different methods for BRDF corrections were implemented based on the literature survey. Table 2 presents a comparison of the BRDF correction methods based on the main approach, dynamic, the use of high-resolution (HR) normalized difference vegetation index (NDVI), use of moderate resolution (MR), inclusion of land cover classification and the coefficients at HR. The High Resolution Adjusted BRDF Algorithm (HABA) is the only algorithm that estimates the BRDF parameters at S2 spatial resolution including a disaggregation of the BRDF parameters estimated from the MODIS surface reflectance product (M{O,Y}D09) at 0.01° spatial resolution and using the Vermote Justice Bréon (VJB) method [38]. A global method based on a static set of coefficients issued in Roy et al. [39,40] and a scene dependent set based on BRDF dynamic characterization [41], namely HABA, was implemented here.

**Table 2.** A summary of the high-resolution BRDF-adjustment algorithms.

| Algorithm | BRDF Approach | BRDF Dynamic | Use of HR NDVI | Use of MR BRDF | Land Cover | BRDF Coefficients at HR |
|---|---|---|---|---|---|---|
| C-factor [1] [39,40] | MCD43 | No BRDF dynamic | No | No | No | No |
| LUM [42] | MCD43 | spatial and temporal variations | No | MCD43 at 500 m | Yes (CDL[2]) | Yes (but per crop) |
| VI-dis [43] | VJB | Spatial and temporal variation | Yes | MODIS VJB at 1250 m | No | No |
| HABA [41] | VJB | Spatial and temporal variation | Yes | MODIS VJB at 1000 m | Yes | Yes |

[1] Current HLS BRDF normalization. [2] CDL: Cropland Data Layer [44].

The two methods were selected to fulfill the accuracy requirements of users. The first approach proposed by Roy et al. [39,40] is widely adopted in the community to correct S2/LS8 data (observations performed close to nadir) due to its simplicity and relatively good performance depending on the land cover. The results are in general more accurate with the HABA, which provides HR characterization, and the correction is performed at the S2 pixel level. However, the HABA implementation is more complex and requires more processing resources.

As shown in Figure 4, the two methods were implemented within the same configurable processing. Regarding the normalization step specifically, there are two major differences between the two methods. On one hand, Roy et al. [39] provides a single set of coefficients. On the other hand, the HABA provides per pixel values, estimated by using the NDVI parameter as proxy value. Furthermore, HABA requires as input a MGRS tile-dependent BRDF characterization.

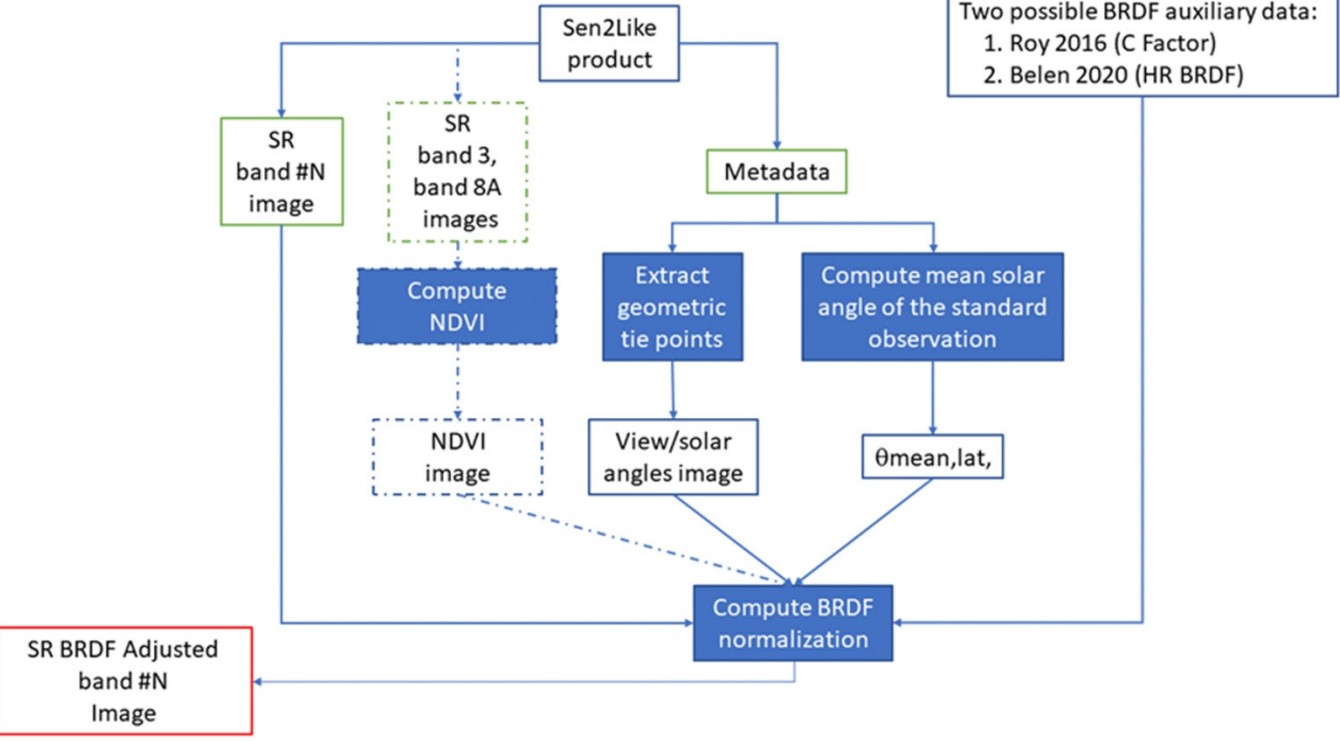

**Figure 4.** S2L BRDF correction workflow.

In the implementation, the BOA reflectance ($R$) is normalized at Nadir, $R^N$, by using the following mathematical relationship (Equation (1)):

$$R^N(\theta_{mean,lat}, 0, 0, \Delta) = R(\theta, v, \phi, \Delta) \times \frac{1 + V(\Delta)K_{vol}(\theta_{mean,lat}, 0, 0) + R(\Delta)K_{geo}(\theta_{mean,lat}, 0, 0)}{1 + V(\Delta)K_{vol}(\theta, v, \phi) + R(\Delta)K_{geo}(\theta, v, \phi)} \quad (1)$$

where

- $K_{vol}$, $K_{geo}$ ([45–47]) are derived based on the pixel-based solar and view angles,
- $V$ and $R$ are the volume and the roughness parameters that describe the BRDF shape,
- The $\theta, v, \phi$ parameters are the solar/viewing, zenith and the relative azimuth,
- The $\Delta$ parameter is related to the spectral band taken into consideration for this calculation.

In Equation (1), the BRDF normalization is performed for a fixed solar zenith angle (SZA) value, $\theta_{mean,lat}$. The $\theta_{mean,lat}$ parameter changes depending on the MGRS tile because it is a function (sixth degree polynomial fit) of the MGRS tile scene center latitude, as listed in Equation (2). It is worth noting that this parameter does not vary with time. The polynomial coefficients used to retrieve the mean solar angle as a function of the MGRS tile latitude (NASA HLS, [48]) are given in Table 3.

$$\theta_{mean,lat} = k_6 \times lat^6 + k_5 \times lat^5 + k_4 \times lat^4 + k_3 \times lat^3 + k_2 \times lat^2 + k_1 \times lat^1 + k_0 \quad (2)$$

where:

- $k_n$ are six-degree polynomial coefficients—the parameter values are listed in Table 3,
- $lat$ is the latitude of the MGRS tile scene center.

**Table 3.** Polynomial coefficient to retrieve the mean solar angle as a function of the MGRS tile latitude (NASA HLS, [48]).

| $k_0$ | $k_1$ | $k_2$ | $k_3$ | $k_4$ | $k_5$ | $k_6$ |
|---|---|---|---|---|---|---|
| 31.0076 | $-0.1272$ | 0.01187 | $2.40 \times 10^{-5}$ | $-9.48 \times 10^{-9}$ | $-1.95 \times 10^{-9}$ | $6.15 \times 10^{-11}$ |

Unlike Roy et al. [39], the HABA algorithm is scene/tile dependent and therefore requires a yearly time series of S2 BOA images as input. The workflow of the BRDF auxiliary data generation is shown in Figure 5.

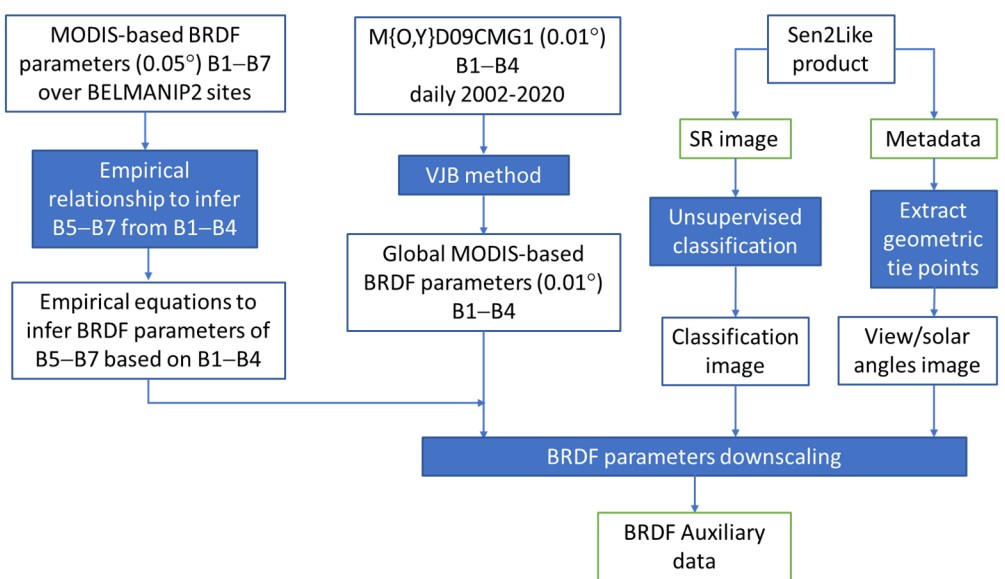

**Figure 5.** Workflow of the BRDF auxiliary data generation.

The *V* and *R* parameters in the HABA are retrieved based on the VJB method applied to MODIS daily directional surface reflectance 0.01° CMG data during the period 2002–2006, 2019 and 2020, assuming that

$$\frac{f_{vol}}{f_{iso}} = V = V_0 + V_1 NDVI \tag{3}$$

$$\frac{f_{geo}}{f_{iso}} = R = R_0 + R_1 NDVI \tag{4}$$

Note that the outputs are the global scale BRDF parameters $V_0$, $V_1$, $R_0$ and $R_1$. The HABA is based on downscaling 0.01° BRDF coefficients to S2L resolution, considering as a reference 20 classes obtained from the unsupervised classification of S2L images and assuming that the surface reflectance of a 0.01° pixel can be written as a weighted sum of *n* number of S2L classes [49].

Given that the M{O,Y}D09CMG product is limited to MODIS B1–B4, we need to derive the BRDF parameters for B6-B7, which are also needed to normalize S2L. Following Villaescusa-Nadal et al. [50], we use MODIS CMG data at 0.05° (which includes all MODIS bands) to obtain a physical relationship between the BRDF parameters for B6–B7. To do this, we first extracted the surface reflectance data for individual pixels in 445 Benchmark Land Multisite Analysis and Intercomparison of Products (BELMANIP2) sites [51]. The BELMANIP2 sites are an update of the BELMANIP1 and were selected due to their representativeness and variability of vegetation types and climatological conditions around the world. Second, the surface reflectance data time series (2002–2020) were sorted in ascending values of the NDVI, which were divided into five groups, using the 20th, 40th, 60th and 80th percentiles as group edge values. Third, we extracted the V and R values of each group and spectral band. This means that for every pixel and every band, we obtained five different V and R values. When using all the BELMANIP2 sites, we obtained a total of 2225 points (445 × 5) for each band. Finally, we applied simple regression between the obtained parameters for the VNIR bands and with the NDVI itself, which is summarized in Equation (5).

$$\begin{aligned} V_{B6} &= 0.55 \times V_{B1} + 0.57 \times V_{B2} - 0.26 \times V_{B3} - 0.11 \times V_{B4} \\ V_{B7} &= 0.95 \times V_{B1} - 0.37 \times V_{B3} \\ R_{B6} &= 0.47 \times R_{B1} + 0.61 \times R_{B2} \\ R_{B7} &= 0.91 \times R_{B1} + 0.34 \times R_{B2} \end{aligned} \tag{5}$$

The method was evaluated by assessing the impact of the approach on the normalized reflectance. To do so, in Figure 6, we compare the time series of noise (2002–2020) of the directional reflectance (raw) with the normalized reflectance using the VJB method in HABA and using the model prediction. The results show that the VJB parameters and the predicted parameters provide a very similar reduction of noise.

The unmixing of BRDF parameters is inferred individually from each image. However, this approach may generate noise in the BRDF inversion, mainly due to clouds, as they reduce the number of cloud-free pixels utilized in the BRDF inversion and thus limit the number of observations available for inversion. Additionally, the low quality of the cloud mask may also introduce noise in the BRDF unmixing. Therefore, based on the individual BRDF unmixing of each image for a given time period, we apply a linear regression for each pixel of V and R parameters versus the NDVI to derive the $V_0$, $V_1$, $R_0$ and $R_1$ parameters at S2L, which are stored as BRDF auxiliary data as the final output of this module.

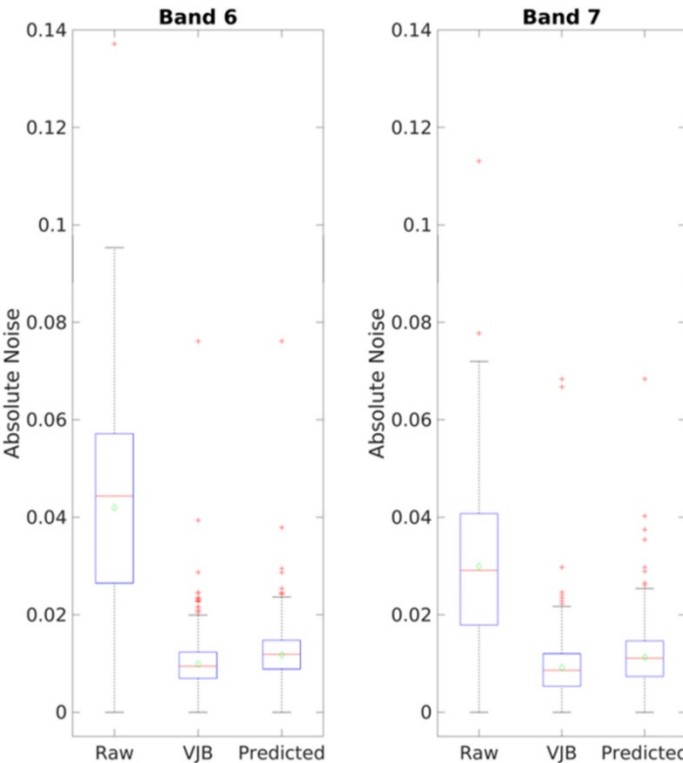

**Figure 6.** Time series evaluation of noise of the directional reflectance (raw) with the normalized reflectance using the VJB method and using the predicted equations. The top line of each box is the upper quartile (Q3) and the bottom line of the box is the lower quartile (Q1). The central line and the green point represent the median and the average values, respectively. The top/down whiskers denote the upper/lower adjacent value. The red crosses depict the values, which fall outside the lower/upper adjacent value interval.

### 2.3.4. Spectral Band Adjustment Factor (SBAF)

Even when the spectral band definitions of the optical instruments under consideration are very similar, the differences in the spectral response can cause significant divergence between the TOA reflectances sensed by them. In Teillet et al. [52], the definition of SBAF is given. The SBAF is target specific and requires hyperspectral data to build a spectral shape over the visible/NIR/SWIR wavelength domain of the electromagnetic spectrum. The OLI/MSI reflectances are predicted based on the spectral response of the respective sensor. The OLI/MSI SBAF coefficient set for the data observed over three pseudo invariant calibration sites (PICS) was computed by South Dakota University [53], which also showed that the SBAF over the desert is mostly independent of the location and season.

Rather than considering a single land cover, the NASA HLS team addressed the SBAF issue [8] by analyzing MSI/OLI SBAF variability for different biome types. So far, a total of 160 globally distributed Hyperion hyperspectral scenes have been processed to SR, and 500 spectra were analyzed. The study revealed the importance of discerning the differences sourced from the spectral calibration from those coming from the radiometric calibration. Furthermore, it also showed that in the context of S2/LS8, the residual errors associated to SBAF adjustment make a very small contribution in the overall error budget.

In addition, S2L adjusts LS8, LS9 and S2B reflectances to S2A reflectance by using univariate linear regression and a single set of coefficients per band. The band-dependent linear transformation factors (slope, intercept) given in [48] for distinguishing S2A and S2B data were used here.

### 2.4. Validity Mask

The objective of the S2L per pixel validity mask is to support the multi-temporal analysis of the L2H/L2F dataset by flagging invalid pixels caused by clouds, cloud shadows, image processing artifacts, etc. In general, it is expected that such quality assurance information is already available within the L1/L2 products. However, depending on the product version and the data provider, the reliability and completeness of this information is varying. For instance, the accuracies of S2 L1C and L2A validity masks are different as depicted in Figure 7. To mitigate the inconsistencies, a morphological operator (dilation) is applied to S2 L1C quality masks in order to increase their quality by excluding the cloud shadow and haze (Figure 7c) from a valid pixel sample size.

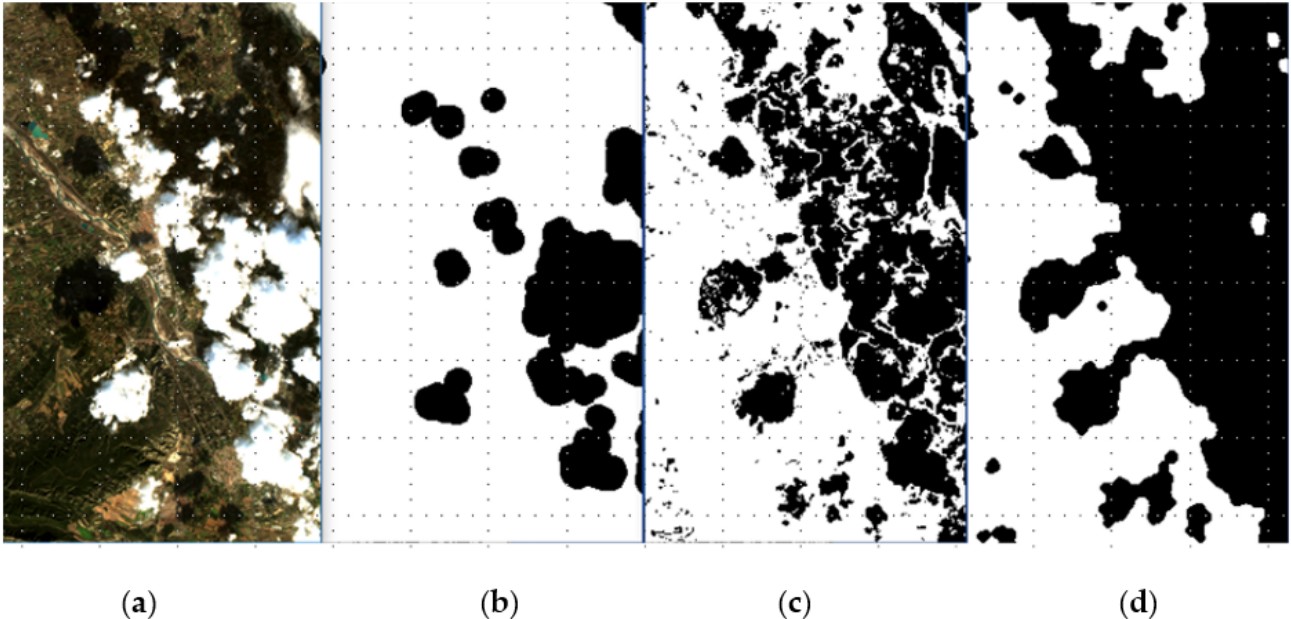

**Figure 7.** Differences between the S2 L1C and L2A cloud masks: (**a**) RGB image, (**b**) L1C cloud mask (inverted), (**c**) L2A cloud and shadow mask and (**d**) post-processed cloud and shadow mask (cleaned and dilated).

For Landsat products, the validity mask is constructed from the LS8/LS9 Band Quality Assessment as described in [54,55], whereas for S2 products, the validity mask is constructed from the L1C cloud mask (either in vector or raster format) and from L2A Scene Classification (SCL) map [56]. Figure 8 illustrates the L2A SCL used to construct the S2L validity mask. While the vegetated and non-vegetated pixels are considered as valid, the snow, cirrus, cloud, cloud shadows and topographic shadows are considered as invalid. The SCL classification performance is regularly reported in L2A Data Quality Reports [13] as well as in independent intercomparison exercises such as CMIX [57].

In order to report on the impact of the validity mask in the data fusion, an additional fusion validity mask is embedded within the L2F product, which flags pixels for which the upscaling algorithm exhibits excessive residual errors (above 20%).

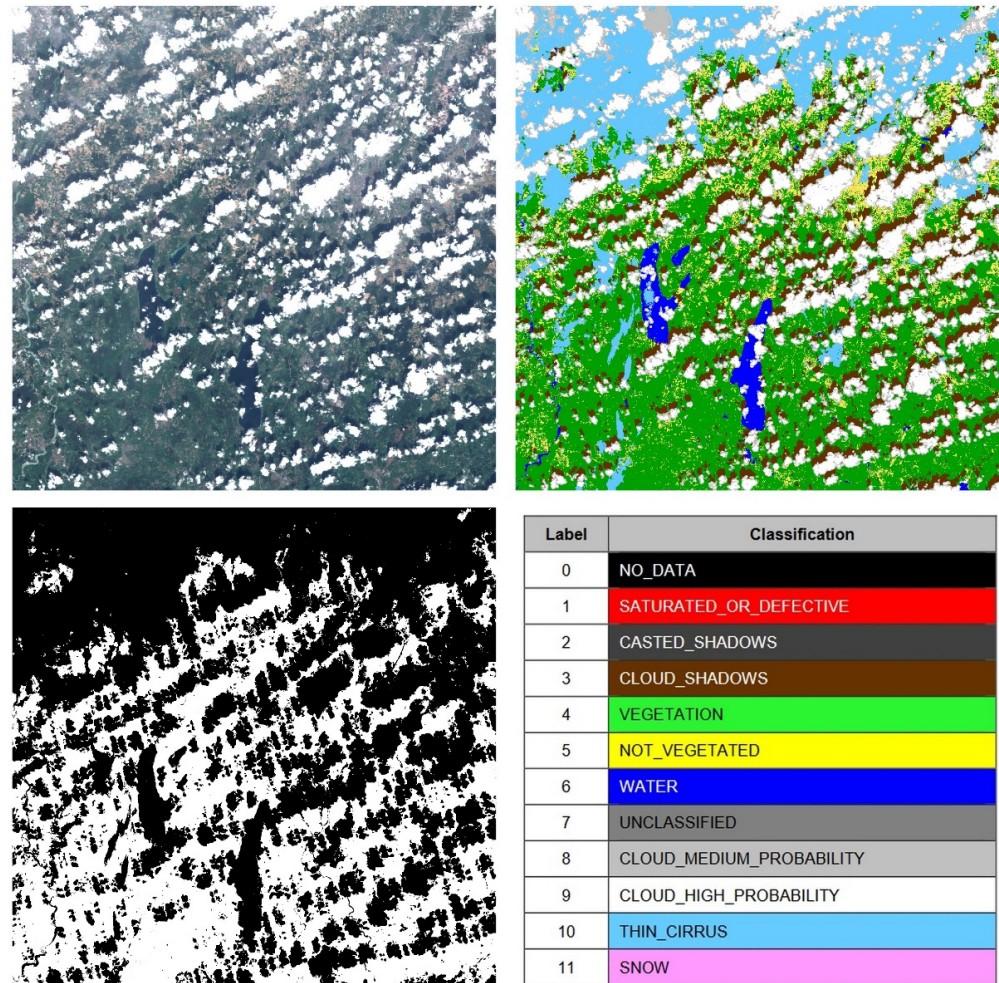

**Figure 8. Top Left**: L1C True Colour Image, **Top Right**: L2A SCL Map. **Bottom Left**: Sen2like Validity Mask, **Bottom Right**: L2A SCL Legend.

### 2.5. Data Fusion

The main objective of the S2L data fusion is to produce data similar to S2 at the observation date/time of LS8/LS9 observations. The approach is based on improving the spatial resolution of LS8/LS9 images from 30 m to 10 m in the case of VNIR bands while preserving radiometric inter-calibration between sensors. Increasing the temporal frequency of S2 HR data is fundamental for many applications dealing with the mapping of rapid changes, as discussed in [58]. The study reported a fusion method to generate Landsat-like images by using MODIS data. Besides this, in the context of monitoring, developing a fusion method for enabling change detection in a timely manner is of great importance for a near real time alert system. The near real time approach requires the improvement of the spatial resolution of a given LS8/LS9 image by exclusively using S2 images observed in the past.

Several spatiotemporal data fusion methods have been developed in the literature to produce synthesized images with both high spatial and temporal resolutions from two types of satellite images: frequent coarse-resolution images and sparse fine-resolution images ([59,60]). The proposed methods are in most cases time consuming and are not tailored to the needs of our study, as we use sparse medium-resolution images (LS8/LS9 data) and fine-resolution images (S2A/S2B data) with higher temporal resolution.

The current S2L approach is mainly based on conventional fusion methods, which aim at combining the HR panchromatic band and low-resolution MS bands from the same sensor and at the same time. However, the S2L methodology slightly differs from the

conventional methods in terms of (i) its multi source environment, (ii) the requirement to preserve calibration of temporal information and (iii) the temporal difference between the data.

The EO satellites sense different characteristics of a landscape, which can be basically categorized as large and small-scale features attributed to low and high spatial frequency content, respectively. From the image processing point of view, the large-scale features are regular, not necessarily uniform, with no discontinuity, whilst the small-scale features are associated with contour and texture. The S2L approach relies on this basic decomposition to improve the spatial resolution of LS8 at a given date. The LS8 large-scale features are complemented with predicted S2 small-scale features and small-scale features not captured by LS8, as described with the following equation.

$$L8^{10m} = S_{L8}^{30 \to 10} + D_{S2-L8}^{10m} \qquad (6)$$

where:

- $L8^{10m}$ is the final LS8 image at the S2 spatial resolution, with deconvolution from 30 m to 10 m,
- $S_{L8}^{30 \to 10}$ is the original LS8 image resampled from 30 m to 10 m by using bilinear interpolation, which is associated to the phase of signal,
- $D_{S2-L8}^{10m}$ is the image of differences derived between 30 m and 10 m spatial resolution predicted by using S2 data and associated with the amplitude of the signal.

The essence of this method is the computation of images of differences. The prediction is performed based on the historical S2 information assuming a linear relationship between the observations. A linear regression model with least squares fitting is used to estimate the transformation parameters. The selection of multi date pixels in the S2 past dataset is based on validity mask information, which needs to be reliable to obtain high performance.

## 3. Validation Results

The geometric accuracy performances of the S2 and LS8/9 L1C/L2A products are well known. The S2 Mission Performance Center (MPC) presents the status of products monthly in the data quality reports ([13]). The NASA science team releases also regular reports. However, an example from the geometric accuracy results, before and after the application of S2L co-registration, is shown in Figure 9. The geometric co-registration of images with respect to a raster reference is demonstrated with two graphics: circular error plots (left) and multi-temporal plots (right). A three year period dataset observed in 31TFJ MGRS tile was used as reference. The upper graphics are related to the initial geometric accuracy (relative to raster reference) and the lower plots are related to the final S2L accuracy in the figure. The precision of the S2A/S2B images varies since only the most recent products were co-registered to the S2 GRI. The LS8/LS9 images were from Collection 1 and Collection 2 products, respectively. Whilst the geometric accuracies expressed in Circular Error at the 90th percentile (CE90) of LS8/LS9 and S2 input images reached 18.81 m and 7.29 m (Figure 9 upper graphics), the S2L correction increased the accuracy values within 2 m (CE90).

Regarding the BRDF processing, comparisons between different observations from adjacent swath overpasses to cover different observation geometries and illumination conditions over Belgium during 2019 were carried out (Figure 10). Figure 11 shows the absolute differences for bands B02, B03, B04, B8A, B11 and B12 regarding the directional reflectance (DIR) without BRDF correction, the C-factor correction ([39]) and the HABA correction applied for different view zenith angles (VZAs). The results were obtained from the comparison of 964 million cloud-free pixels for each date. The number of overlapping pixels varies between 9% and 36% for VZAs from 0° to 2° depending on the swath for each MGRS tile and the acquisition date. For VZAs between 2° to 4°, the percentage of overlapping data represented between 28% and 45% of the tiles. Additionally, for VZAs between 4° and 6°, the number of analyzed pixels represented between 13% and 25%.

Finally, for the largest VZA observations (>6°), the analyzed pixels represent between 12% and 42%.

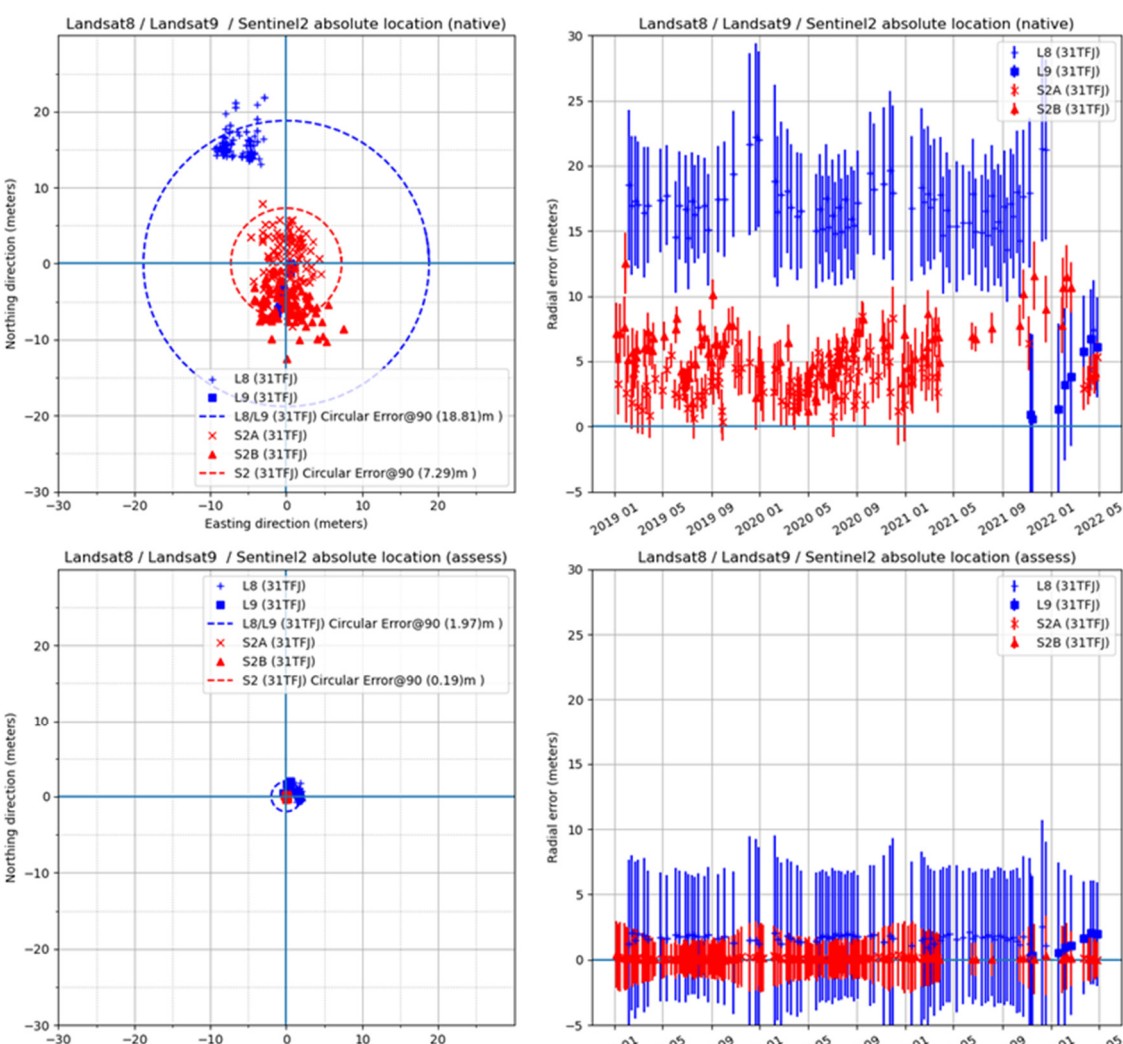

**Figure 9.** Sen2Like geometric processing quality control results.

The separation of intervals of absolute difference in the boxplot (Figure 11) shows that the metrics in the visible spectrum (B02, B03, B04) yielded the lowest errors with a median under 0.01 and a low variability for all methods and the observation geometries. Besides, the impact of the BRDF correction in B8A, B11 and B12 depends on the observation geometry (VZA) and on the BRDF correction method. For VZA lower than 4°, the directional reflectance shows the lowest errors, while C-Factor and HABA show errors up 0.01. For VZA over 6°, the C-Factor stills shows an overestimation in B8A reflectances, and HABA stabilizes the average value but still exhibits a higher variability than the directional reflectance. In the case of the SWIR bands (B11 and B12), the C-Factor and the HABA perform a better correction for the larger VZA acquisitions. The comparison shows that the HABA produced the lowest differences across all bands, while the C-factor correction showed larger discrepancies than the directional product for all VZA pixels aggregated. Table 4 summarizes the metrics obtained from the analysis, demonstrating that the HABA correction decreases the absolute difference up to 13% (in the SWIR bands) compared to the directional product.

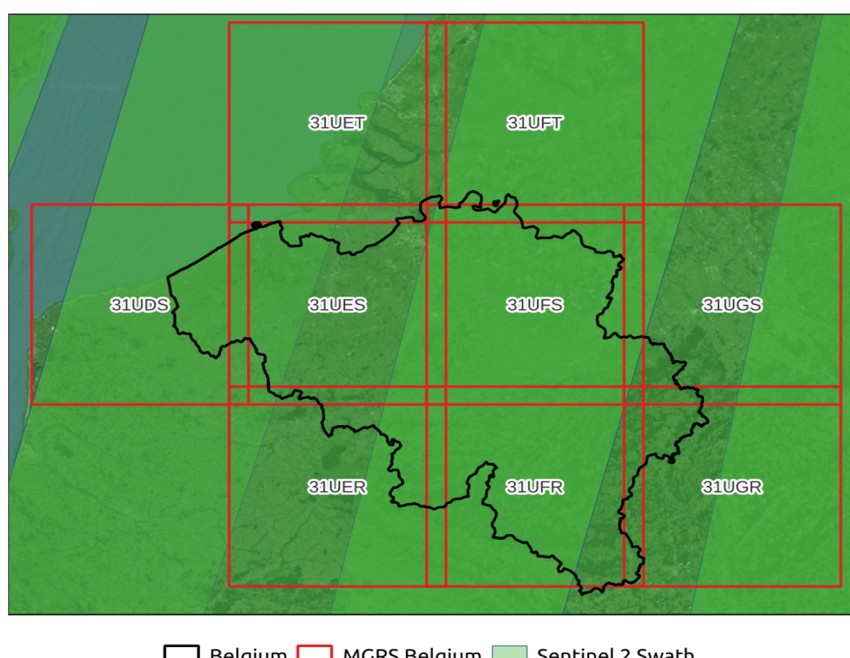

**Figure 10.** BRDF evaluation area over Belgium. Red lines show the MGRS tiles, and green polygons show the S2 swath coverage. Darker green areas show overlaps between neighboring swaths.

**Table 4.** Summary of difference of S2L bands over Belgium MGRS tiles, considering a window of 1 to 5 days. Columns shown are the mean, standard deviation and correction percentage of NBAR algorithms. Positive and negative values represent increase and decrease of errors, respectively.

| Band | Method | Mean | Std | % Corr |
|------|--------|------|-----|--------|
| B02 | DIR | 0.01297 | 0.01174 | |
| | C-FACTOR | 0.01482 | 0.01402 | 14.24% |
| | HABA | 0.01213 | 0.01139 | −6.45% |
| B03 | DIR | 0.01179 | 0.01207 | |
| | C-FACTOR | 0.01397 | 0.01460 | 18.56% |
| | HABA | 0.01062 | 0.01185 | −9.90% |
| B04 | DIR | 0.01144 | 0.01318 | |
| | C-FACTOR | 0.01357 | 0.01552 | 18.58% |
| | HABA | 0.01051 | 0.01279 | −8.19% |
| B8A | DIR | 0.01994 | 0.02038 | |
| | C-FACTOR | 0.02431 | 0.02213 | 21.96% |
| | HABA | 0.01742 | 0.02002 | −12.60% |
| B11 | DIR | 0.01424 | 0.01842 | |
| | C-FACTOR | 0.01753 | 0.02012 | 23.03% |
| | HABA | 0.01243 | 0.01791 | −12.74% |
| B12 | DIR | 0.02621 | 0.01686 | |
| | C-FACTOR | 0.03134 | 0.01851 | 19.58% |
| | HABA | 0.02442 | 0.01614 | −6.82% |

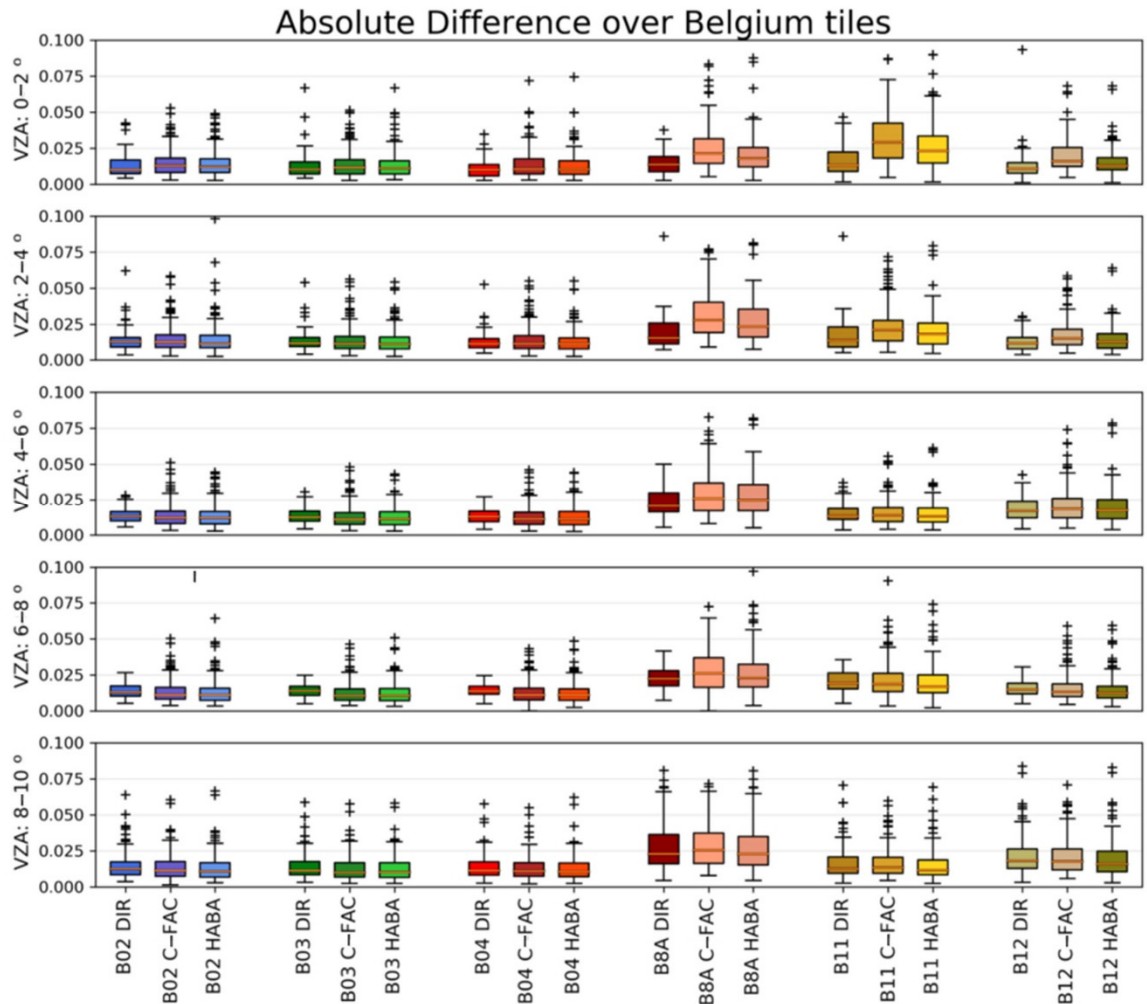

**Figure 11.** Absolute VZA differences between S2L bands over Belgium MGRS tiles.

As shown in the Table, the BRDF processing with the HABA brings substantial improvements regarding time series smoothness. However, as discussed before, this methodology is scene dependent. In addition, from the on-demand production perspective, it is not straightforward to argue that the HABA results systematically reach operational goals for any newly generated dataset. The Temporal Smoothness Index (TSI) is widely used in the community in order to assess the consistency of the spatiotemporal dataset, e.g., in [8]. Experience has revealed that the usefulness of the TSI might be limited to discriminating algorithm improvements and comparing time series together. For this reason, it is proposed to complement the TSI with a quality control metric defined as a correlation measurement between two samples, samples taken for every image pixel and including temporal information. For a given observation date, the two elements, which are the viewing/sun azimuth difference values and the difference between pixel value and predicted value (as a result of linear interpolation by using the two closest dates), are considered. This approach creates a TSI correlation image reflecting information on the angular dependency of pixel basis temporal information. Furthermore, the ESA Land Cover (LC) Climate Change Initiative (CCI) Land Cover map (Version 2.0.7) with a 300 m resolution [61] is used as supporting data to compute the image statistics over regional LC classes, keeping a valid pixel sample by using the S2L validity mask. The CCI LC map is compared with the TSI correlation extracted from the COG L2H overview images (120 m resolution). The TSI correlation metric results generated for the three methods, i.e., DIR, C-Factor and the HABA for the 31TFJ multi temporal dataset, which includes 113 LS8/S2 observation (44 S2A, 40 S2B, 28 LS8) dates over the full-year 2019 period, are presented in

Figure 12. Based on this experimental TSI correlation metric, it becomes evident that the C-Factor and the HABA methods reduce angular dependencies. Compared to the C-Factor, the HABA performs better in most cases: the correlation results are below 0.01, except for Tree Cover (LCCS Value 60; 70, 90) and Bareland Areas (LCCS Value 200), in particular for S2L bands B03, B04 and B12. In addition, the results for B02 do not show improvements, even if the same tendency regarding the underperformance of HABA for the LC class previously mentioned is observed.

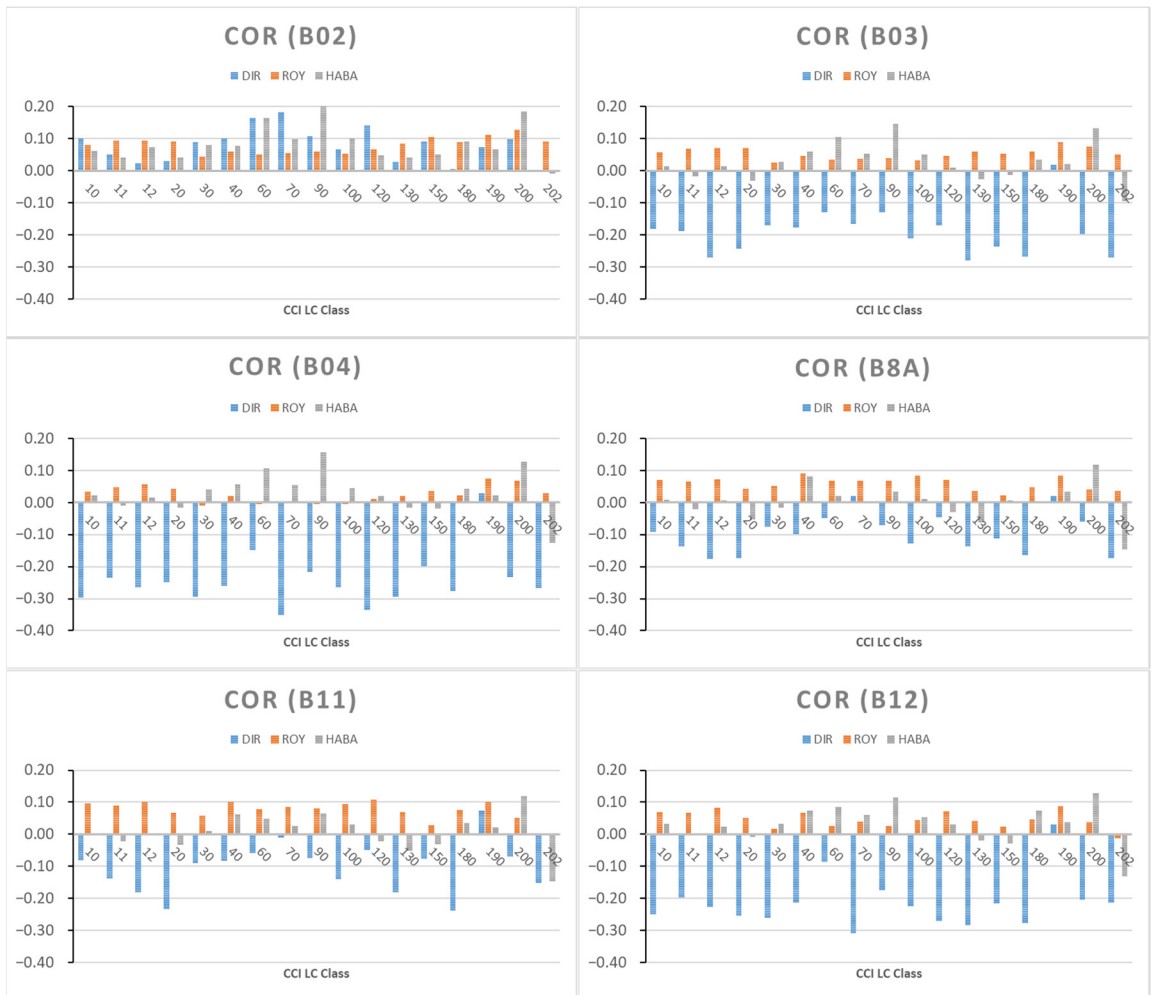

**Figure 12.** TSI Correlation metric results for S2L bands and for each CCI LC class (31TF, year 2019J).

Regarding the data fusion, from a quantitative point of view, we used accuracy precision uncertainty (APU) defined in [62] to compare datasets, as performed in [37]. Thus, higher values of A, P and U reflect higher discrepancies between datasets, and low values reflect a better agreement. The LS8 L2H and L2F images for three bands (B02, B03, B04) were compared with respect to S2 L2H/L2F images. Data acquired at the same date were selected to eliminate the temporal differences. The results given in Table 5 show that the fusion algorithm does not modify the inter-calibration between LS8 and S2A (LS8/S2A) significantly, and the results are in agreement with the cross-calibration results given in [53]. Moreover, this shows that the inter-calibration does not change depending on the S2L processing level, i.e., L2H and L2F.

**Table 5.** Validation of S2L fusion algorithms, image comparison between LS8 L2H, LS8 L2F and S2A image (as reference) dated 14 July 2019, tile 12SVB).

| | | Pixel Number | Mean S2A | Mean LS8 | LS8/S2A | Accuracy (A) | Precision (P) | Uncertainty (U) |
|---|---|---|---|---|---|---|---|---|
| L2H (30 m) | B02 | 99,346 | 0.109 | 0.103 | 0.945 | 0.006 | 0.008 | 0.010 |
| | B03 | 99,358 | 0.149 | 0.148 | 0.996 | 0.001 | 0.009 | 0.009 |
| | B04 | 99,321 | 0.205 | 0.201 | 0.980 | 0.004 | 0.012 | 0.013 |
| L2F (10 m) | B02 | 99,159 | 0.109 | 0.103 | 0.944 | 0.006 | 0.007 | 0.009 |
| | B03 | 99,208 | 0.149 | 0.148 | 0.996 | 0.001 | 0.008 | 0.008 |
| | B04 | 99,380 | 0.205 | 0.201 | 0.980 | 0.004 | 0.010 | 0.011 |

As the LS8 L2F images are synthetic data, this enables the retrieval of information over ROIs that are not available in native LS8 images such as nearby crop field parcel limits. Many pan-sharpening techniques increase the image contrast with the risk of altering pixel values on both sides of the image edge, causing edge artefacts ("halos"). To quantify this effect, nine ROIs with different geometric properties and high spatial frequencies have been selected, as shown in Appendix B Figure A2. The LS8/S2 L2H/L2F mean edge profiles were computed, and an accuracy analysis and visual check were performed (Appendix B Figure A3). As shown in Figure 13, the LS8/S2 L2F uncertainties do not exceed 0.02 (BOA) in the evaluated bands.

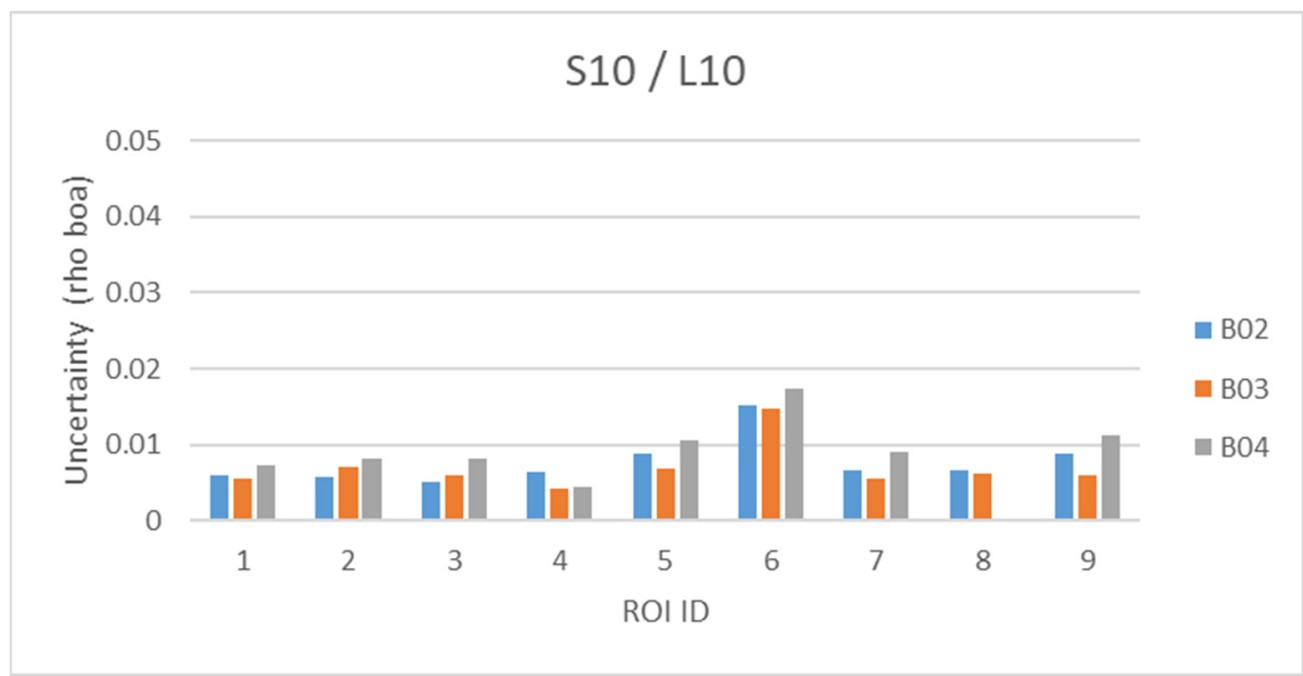

**Figure 13.** LS8/S2 L2F uncertainties for the nine ROIs.

The original LS8 images and the S2-like LS8 30 m images (L2F images) are shown in Figure 14 for visual comparison. The qualitative assessment shows that the S2L deconvolution process preserves the radiometry and improves the object delineations for different land cover types.

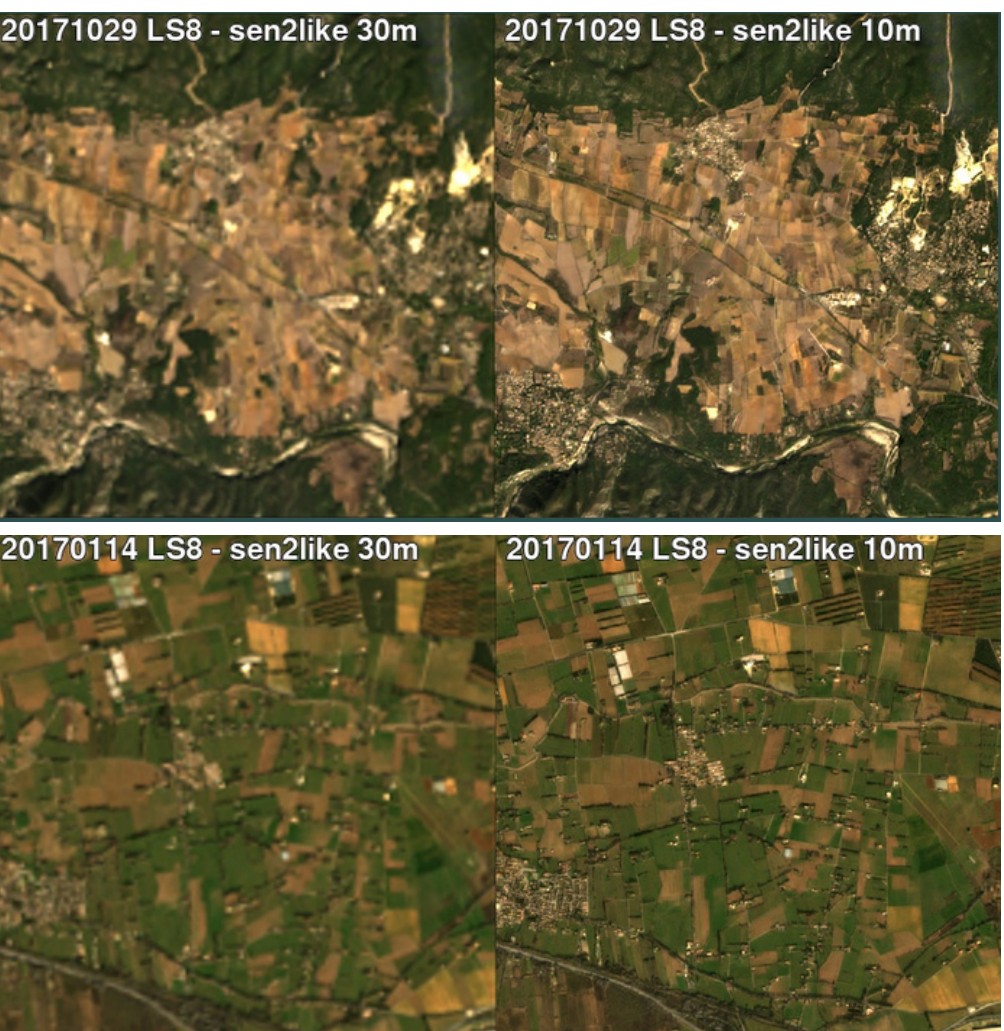

**Figure 14.** Thirty meter LS8 (**left**) and 10 m LS8 L2F (**right**) images over a valley (**above**) and crop fields (**below**).

## 4. Discussions

### 4.1. The S2L Framework

The S2L algorithm allows the harmonization and fusion of multi-sensor optical data for the purpose of ARD preparation with the ultimate goal of generating VCs. The S2L tool developed for this purpose is able to apply geometric processing, spectral inter-calibration and AC, BRDF and spectral adjustments and data fusion to ensure that the input products are at the same spatial resolution. The tool framework is flexible, and it is designed to integrate with the future products of different spaceborne and airborne optical missions with the condition that the proper calibration information is available. The tool development principle also involves reusability from the existing qualified tools such as SMAC, Sen2Cor and qualified methods.

The major achievements of the study include detailed investigations of the quality of a spatiotemporal dataset, algorithm comparison and the development of an ARD quality assurance approach. The validation of a product's radiometric and geometric qualities is fundamental to propose this novel processing strategy for the realization of VCs as a further step in EO data fusion activities. Thanks to S2L processing, data from different sources are corrected, information is created with the generation of high-quality validity masks, and the resulting produced L2F images have the ability to reconstruct vegetation dynamics through NDVI temporal profiles and are robust to temporal change and cloud cover. The S2L is focused on physical quantities with the potential to progress in uncertainty analysis. In

addition, a major limitation is that the approach depends on the information level disclosed by the data provider and also on the intrinsic quality of the input mission—the product format design, calibration monitoring and product processing quality.

### 4.2. Adaptation to Further Optical Missions

The S2L software offers a generic framework and enables the creation of a multi mission seamless dataset. The VC of the S2A/B and LS8/9 sensors already provides a global median average revisit of 2.3 days [3]. The inclusion of further missions increases the data availability, shortens the temporal revisit interval and maximizes cloud-free observations. Within this scope, the integration efforts to include further satellites with similar characteristics as well as hyperspectral missions and data obtained from airborne platforms including the UAVs are addressed in this section.

It has previously been shown that the integration of Indian Remote Sensing (IRS)-ResourceSat-2 [63] in S2L requires minor development efforts, and the most critical issues remain in the knowledge of sensor radiometric and spectral calibration for which documentation is not disclosed. The lack of calibration information would yield inaccurate atmospheric correction results and cause difficulties in spectral adjustments.

On the other hand, the studies performed on the Deutsche Zentrum für Luft und Raumfahrt (DLR) Earth Sensing Imaging Spectrometers (DESIS) [64] yielded promising results on the integration of hyperspectral data into S2L by aggregating the TOA images. The aggregation process consists of convoluting the TOA hyperspectral VNIR spectra from DESIS with response functions of the S2 MSI sensor and adding the SWIR bands of the original S2 TOA values [65]. The approach allows researchers to avoid changes in the atmospheric correction baseline while preserving the data quality (Figure 15) and calibration accuracy. In the approach, the VNIR DESIS bands are resampled from the DESIS spatial resolution (30 m) to the 20 m resolution of S2. The final aggregated TOA dataset can be processed with two different methods. The first method relies on the processing of the dataset in the same way as DESIS data relying only on VNIR bands (DESIS-S2-VNIR). The second method involves processing like S2 using the SWIR-bands (DESIS-S2) as well. Both methods provide similar results, showing no significant differences (Figure 15). The missing SWIR bands of the DESIS sensor do not generate artefacts for this study due to differences in processing. The aggregation of hyperspectral data to MS data on L1C Level opens up new perspectives for the integration of current (PRISMA [66], EnMAP [67]) and future hyperspectral (CHIME, [68]) missions into the MS framework. The main drawback of this approach is that it may not be possible to utilize the full potential of the hyperspectral data with the fusion process.

The S2L preliminary study showed no strong differences in the BOA results between MS and hyperspectral data processing. However, the AC of hyperspectral data is able to use much more information than for the MS data. One example is the availability of shorter wavelength bands with more information for all the algorithms involving analysis in the deep blue wavelengths (e.g., scale path radiance), which is clearly visible in Figure 16. The availability of bands in other water vapor regions such as 820 nm improves the accuracy of the water vapor estimation. In addition, the availability of multiple bands in this and the other water vapor absorption regions such as 945 nm decreases the water vapor uncertainty, although such uncertainty does not have a large impact on the final BOA reflectance.

The convolution of hyperspectral BOA reflectance data to MS has several ad-vantages over aggregating hyperspectral data to MS at L1C for processing. The BOA spectra are smoother than the TOA spectra, which includes absorption features. From the computational and storage point of view, only one hyperspectral product will be stored, serving both the S2L and the hyperspectral L2A users, and fewer computation-al resources will be required to convolve the hyperspectral L2A products to any other MS missions, even when the SRFs of MS missions change. These results also confirm that new MS sensors with more spectral bands than S2 can be easily included in the algorithm (e.g., S2 NG [68], L9 [69]).

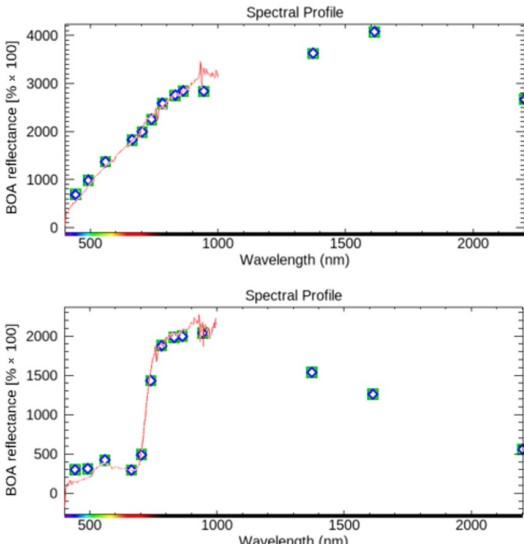

**Figure 15.** BOA reflectance (%) of a soil (a) and a vegetation (b) pixel for hyperspectral DESIS processing (line, red), multispectral DESIS-S2-VNIR processing (diamond, blue) and MS DESIS-S2 processing (square, green). The spectra correspond to a mean over 2 × 2 pixels (upper) and 3 × 3 pixels (lower) to integrate the same geographical area.

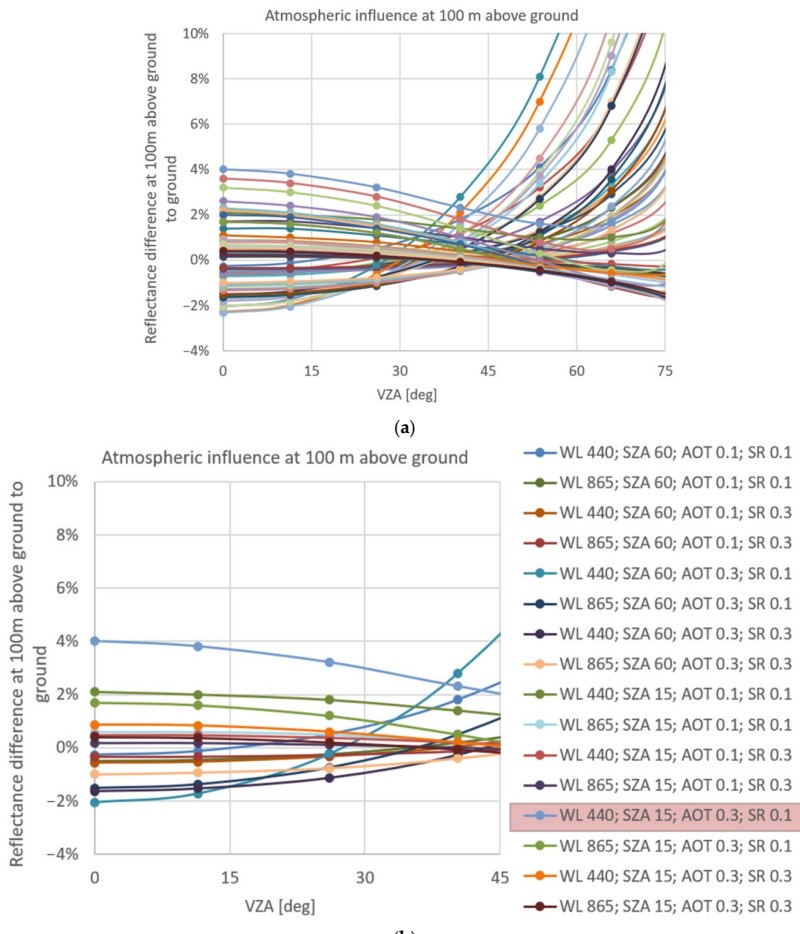

**Figure 16.** Reflectance differences 100 m above ground to surface reflectance on ground for different combinations of inputs to radiative transfer model: surface reflectance 0.1 and 0.3, solar zenith angle 15° and 60° and AOT 0.1 and 0.3. (**a**) overall view, (**b**) enlarged view over VZA between 0°–45°.

Concerning the airborne data streams, the AC of such data is not recommended. As shown in Figure 16, the reflectance recorded from a UAV flying at an altitude of 100 m above ground differs less than 4% from the reflectance at surface for observation angles $\leq 45°$ in the worst case. The atmospheric influence on the UAV data is smaller than the current uncertainty of SR retrieval in AC. The SR retrieval with Sen2Cor is frequently outside the uncertainty goal of about 5% [13].

## 5. Conclusions

In this study, the algorithms and the implementation framework of Sen2Like (S2L), which is an open-source tool initially developed for the harmonization and fusion of S2 and LS8 data to produce Analysis Ready Data (ARD), are presented and discussed. Several issues such as geometric processing, AC, cloud mask, BRDF processing, etc., in S2 and LS8/LS9 data are addressed in the paper. The S2L tool supports several processing steps dedicated to data harmonization and data fusion with respect to the S2 mission as a reference. The S2L algorithm is able to work and adapt in different domains, such as spatial, spectral and temporal. As the airborne and spaceborne remote sensing instruments have different characteristics, the adaptation of the processing steps per sensor is essential in order to provide a consistent and accurate multi-temporal data stack and to enable the generation of dense reflectance time series.

The validation of the ARD shows that the quality of input data is preserved and the inter-calibration between different sensors is improved. The validation activities provided the opportunity to characterize the dataset's consistency and accuracy and demonstrated that the quality information should be included as part of the delivered products. Improved quality information facilitates the use of multi-temporal data.

By providing dense time series at fine spatial scale, the S2L tool aims to be helpful for the fine modeling of change and close monitoring with the integration of new missions. The study also concludes that AC should be mission-specific. Based on the preliminary analyses with the hyperspectral data obtained from the DESIS, it was proven that the S2L framework is able to process the products of any other MS or hyperspectral optical data at the L2A level.

The S2L is currently entering its pre-operational phase and will be offered to the EO community as an "on demand" processor. Besides taking the steps for the transition to the operational phase, the S2L project team is pursuing research and development activities to densify time series for the characterization and measurement of environmental parameters. In this context, research questions and issues related to the fusion of current S2L data with the data from the other sensors and platforms, such as hyperspectral sensors, UAV camera data, etc., are under investigation.

In the era of artificial intelligence, S2L also has the potential to improve the model training processes by providing radiometrically consistent spatiotemporal information and might be helpful for many applications in the field (e.g., for the AI4EO Initiative). As an example, the deep learning models implemented for the LULC classification and employing multi-mission datasets can benefit from the increased data quality with high image radiometric coherence. A newly launched study on the Worldstrat dataset [70] reported that an increase in revisit times of Sentinel-2 images would be highly beneficial for this purpose. It can be envisioned that this requirement can be fulfilled with VCs to some extent.

**Author Contributions:** Conceptualization; S.S., J.L. and V.D.; methodology; S.S., J.L. and V.D.; software, V.D., J.L. and S.S.; validation, S.S., J.L., V.D., B.P., R.D.L.R., B.F. and I.M.L.; formal analysis, S.S., J.L. and E.G.C.; investigation, S.S., J.L., V.D., B.P., R.D.L.R., B.F. and I.M.L.; data curation, S.S., J.L., E.G.C., B.P., B.F. and I.M.L.; writing—original draft preparation, S.S., J.L., S.K., B.P., R.D.L.R.; writing—review and editing, S.S., J.L.,E.G.C., B.F. and S.K.; supervision, S.S., E.G.C., V.B. and F.G.; project administration, E.G.C., V.B. and F.G. All authors have read and agreed to the published version of the manuscript.

**Funding:** This research was funded by European Commission/European Space Agency under Sentinel-2 Mission Performance Center (MPC) Project.

**Data Availability Statement:** The Sen2Like tool is based on free and open-source libraries and is publicly available at https://github.com/senbox-org/sen2like (accessed on 7 July 2022).

**Acknowledgments:** The authors are grateful to the European Space Agency (EOEP5 Block 4) for funding the first stage of this development. We also thank to the anonymous reviewers and the Editors for their insights and valuable contributions to the paper.

**Conflicts of Interest:** The authors declare no conflict of interest.

## Appendix A. S2L Data Description

The Sen2Like products are processed in the SAFE format. Both L2H/L2F product formats are tailored from the S2 product format specifications defined in [18]. As introduced in Figure A1, the S2L L2H/L2F physical format includes in particular (1) the product/tile metadata, (2) preview image, (3) image data and (4) quality indicators described. Besides image data, this section focuses on S2L-specific contents related to the processing-dependent metadata and quality indicator data.

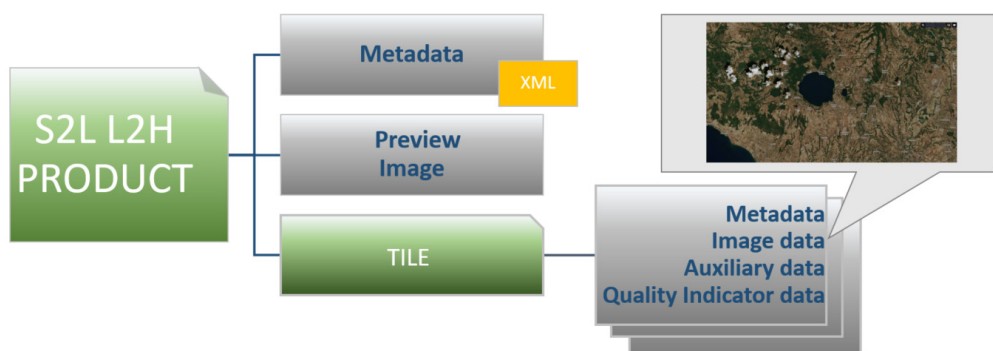

**Figure A1.** Illustration of the Level 2H product physical format (similar to L2F).

*Appendix A.1. Image Data*

The L2H/L2F Image Data (IMG_DATA) folders include harmonized and fused surface reflectance images, respectively. As presented in Table 1, for some S2 bands, there is no equivalent Landsat8 band, and vice versa. These data are kept and stored in a dedicated file directory named the "NATIVE" directory. The content of the Sen2Like Image Data folder depending on the processing level and the mission is listed in Table A1.

**Table A1.** Content of the Sen2Like Image Data folder (when required, the corresponding LS8 equivalent band is labeled with "L" as prefix instead of "B").

|  | Level 2H | Level 2F |
|---|---|---|
| S2 | The harmonized surface reflectance tiles at Sentinel-2 spatial resolution with 7 bands: B01 (60 m), B02 (10 m), B03 (10 m), B04 (10 m), B8A (20 m), B11 (20 m), B12 (20 m) | |
| LS8 | The harmonized surface reflectance tiles with 7 bands at Landsat-8 spatial resolution (B01, B02, B03, B04, B8A (L05), B11 (L06), B12 (L07) | The fused surface reflectance tiles with 6 bands at Sentinel-2 resolution B02, B03, B04, B8A (L05), B11 (L06), B12 (L07) and with 1 band at Landsat-8 resolution B01 (30 m) |
| NATIVES2 | The surface reflectance tiles at Sentinel-2 spatial resolution with 4 channels B05 (20 m), B06 (20 m), B07 (20 m), B08 (10 m) | |
| NATIVELS8 | The panchromatic surface reflectance tile (15 m) and the 2 emissive bands at 30 m resolution (L10 & L11) | |

The L2H/L2F products use the same tiling, encoding and filling structure as the S2 L1C product described in the GMES Space Component—Sentinel-2 Payload Data Ground Segment (PDGS) and Product Definition Document (PDD). In the L2H/L2F raster images, the pixel values are encoded into 16 useful bits, which are directly proportional to SR values. While the standard native GeoTIFF and JPEG 2000 image file formats are supported, it is possible to deliver output image data in the Cloud Optimized GeoTIFF (COG) file format. The COGs' internal overviews are defined according to a pyramidal order. The overview dimensions have been selected to preserve the geometric accuracy and the co-registration of the harmonized datasets at lower zoom levels. Therefore, the following overview resolutions are available for each band, independently of the mission: 20 m, 60 m, 120 m and 360 m. With this approach, it is possible to create reduced resolution spatio-temporal datasets much faster. The COG format, in addition to its internal tiling, is optimized to support Hyper Text Transfer Protocol (HTTP) range requests (Internet Engineering Task Force 2014, https://tools.ietf.org/html/rfc7233, accessed on 7 July 2022). The following configuration options are defined as default: pixel interleaving, internal tiling: 1024, block size: 1024, compression with LZW or Deflate. The resampling method used to create overviews is "average", except for the masks for which the "mode" method is preferred.

In addition to image data, the product includes preview images and quicklook images for which high level specifications are detailed in Table A2.

**Table A2.** Sen2Like preview image data.

| Image Name | Image Format | Resolution | Description |
|---|---|---|---|
| Preview Image | GeoTIFF (8 bit) | 320 m | RGB (3 channels: RED = B4; GREEN = B3; BLUE = B2). Preview dynamic is stretched (min = 0.0, max = 0.250, scale = 255.0) |
| Quicklook Image | JPEG | 30 m | RGB (3 channels: RED = B4; GREEN = B3; BLUE = B2). Preview dynamic is stretched (min = 0.0, max = 0.250, scale = 255.0) |
| Quicklook Image | JPEG | 30m | SWIR-NIR (3 channels: RED = B12; GREEN = B11; BLUE = B8A). Preview dynamic is stretched (min = 0.0, max = 0.40, scale = 255.0) |

*Appendix A.2. Metadata*

For each product and at any processing level, the complete set of metadata is provided as a machine-readable XML file in accordance with the INSPIRE, ISO 19115-2 (Geographic Information-Metadata, Part 2: Extension for imagery and gridded data) and ISO 19119 (Geographic information-Services) standards. As listed in Table A3, the S2L product format contains metadata at two levels: i.e., the product level and the tile level.

**Table A3.** Sen2Like metadata embedded within the L2F and L2H product formats.

| Metadata Name | Description |
|---|---|
| | Product Level |
| Information | • Datatake information: <br> • Datatake unique identifier; <br> • Spacecraft name (Sentinel-2A/B/ … ); <br> • Datatake type (MSI Operation Mode: Nominal, Dark Signal, etc.); <br> • Sensing start time; <br> • Sensing orbit number (relative); <br> • Sensing orbit direction <br><br> • Spacecraft name (Sentinel-2A/B/ … ) |
| | Processing Level <br> List of image files L2H/F composing the products <br> Spectral information (relation between production image channels and on-board spectral bands) <br> Solar irradiance (per band) and the correction U related to the Earth-Sun distance variation <br> Reflectance quantification value (for conversion of digital numbers into reflectance); <br> Special values encoding (e.g., NODATA). |
| Quality Indicator | Cloud coverage assessment |

**Table A3.** *Cont.*

| Metadata Name | Description |
|---|---|
| | Tile Level |
| Metadata | Tile identifier, as referenced by Level-1C data |
| | Tile geocoding: |
| | • Upper-left coordinates (ULX, ULY) of the tile (in m); <br> • Pixel dimensions (XDIM, YDIM) within the tile (in m and depending on band GSD); <br> • Tile size in number of lines/columns. |
| Quality indicator | Mean sun angles (zenith and azimuth) <br> Mean viewing incidence angles per band <br> (zenith and azimuth) <br> Cloudy pixel percentage <br> Pixel level quality indicator: pointer to the QI files <br> Quicklooks and Preview data information: <br> pointer to image files |

*Appendix A.3. Quality Indicator Data*

An additional L2H/L2F Quality Information (QI) report (e.g., L2H_QI_Report.xml) is available within the granule QI_DATA folder (see Table A4). The QI file aggregates all the quality information available in the L2H/L2F processing step. S2L inherits the QI from L2A processing (Atmospheric Correction, Scene Classification) when available. In the QI file, different metrics from the subsequent processing steps (Geometry, BRDF, SBAF, Fusion) are also available. For example, for the geometric step, the following information is available: spectral band of reference, co-registration error (in m) before and after correction, number of points and reference of the method. Furthermore, in the QI file, a place is reserved for Quality Indicator Metadata from auxiliary data used in Sen2like processing (CAMS-ECMWF, GRI, BRDF, VJB).

**Table A4.** Sen2Like QI data and parameters.

| Name | Description | QI Parameters |
|---|---|---|
| L2A_SceneClass | L2A scene classification QI <br> (Sen2Cor version 2.10 ATBD, available at <br> https://step.esa.int/thirdparties/sen2cor/2.10.0 <br> /docs/S2-PDGS-MPC-L2A-ATBD-V2.10.0.pdf <br> accessed on 7 July 2022) | Percentage of classified pixels |
| L2A_AtmCorr | L2A atmospheric correction QI (Sen2Cor version <br> 2.10 ATBD, available at <br> https://step.esa.int/thirdparties/sen2cor/2.10.0 <br> /docs/S2-PDGS-MPC-L2A-ATBD-V2.10.0.pdf <br> accessed on 7 July 2022) | Average values of atmospheric <br> parameters (ozone, water vapor, aerosol) <br> Average solar zenith angle |
| Auxiliary Data | Digital Elevation Model QI, Meteorological data QI | Place reserved, intended for Sen2like 4.2 |
| L2{H,F}_Geometry | Reference of the method (string) <br> QI derived from the geometric assessment <br> processing | BAND, COREGISTRATION <br> BEFORE_CORRECTION, <br> COREGISTRATION <br> AFTER_CORRECTION, <br> NB_OF_POINTS, MEAN, STD, RMSE, <br> SKEW, KURTOSIS [22–25] |
| L2{H,F}_BRDF_NBAR | Reference of the method (string) <br> QI derived from the BRDF processing | KERNEL_DEFINITION, CONSTANT <br> SOLAR_ZENITH_ANGLE, MEAN <br> DELTA_AZIMUTH [38,39] |
| L2{H,F}_SBAF | Reference of the method (string) <br> SBAF coefficients and offsets (values) | SBAF coefficients and offsets per band [8] |
| L2F_FUS L2F only | Reference of the method (string) <br> QI derived from the processing | Sen2like ATBD (unpublished) |

## Appendix B. Validation of Data Fusion Algorithm

As shown in Figure A2, nine ROIs located over the 12SVB MGRS tile are chosen in order to evaluate the performance of the S2L data fusion method. Statistics are computed over image rows or image columns, and the direction of the edge profile is indicated with a blue arrow.

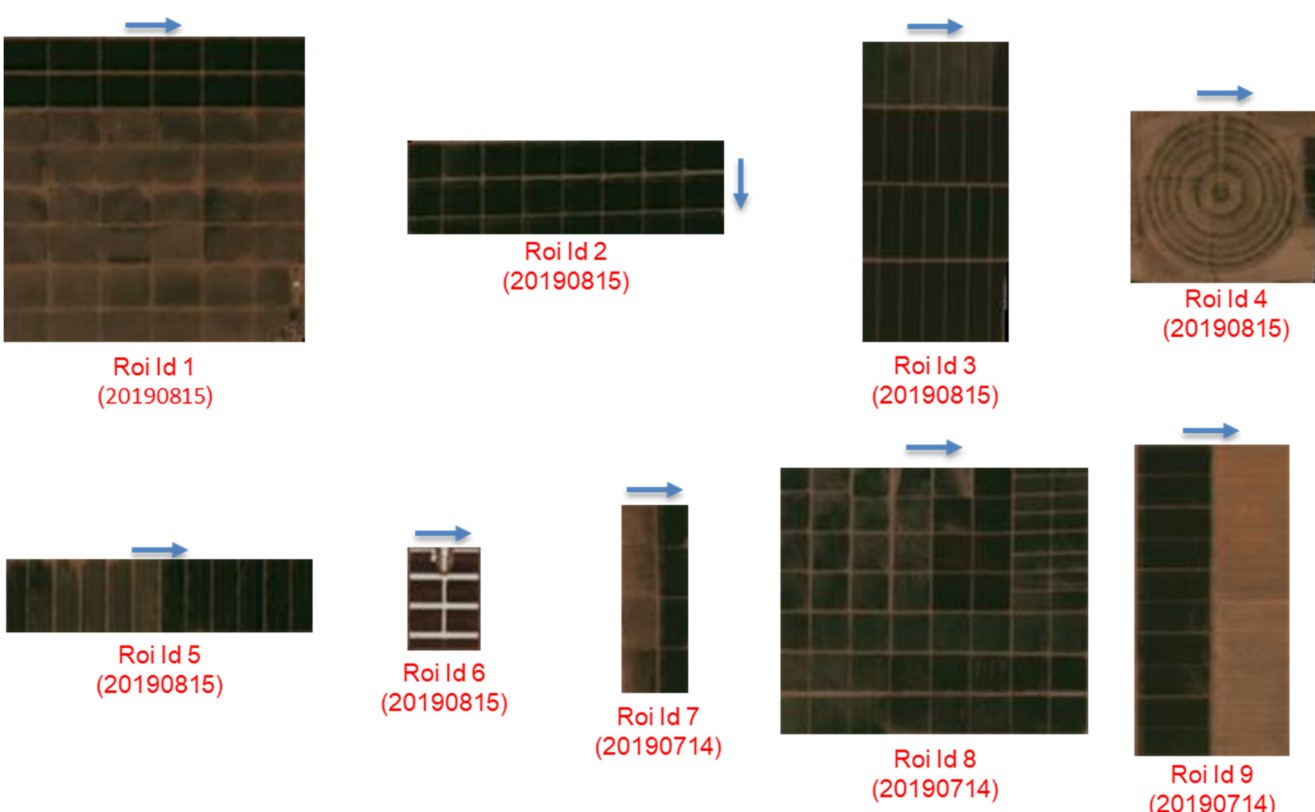

**Figure A2.** Landsat 10.0 m Red Green Blue color composition Full resolution L2F images (10.0 m) of the Nine ROIs selected over the 12SVB MGRS tile. The ROIs are used for validation of data fusion method. The direction of the edge profile is indicated with blue arrows (mean edge profile computed over records in the orthogonal direction), and the ROI identifier and the observation date are indicated in red.

B02

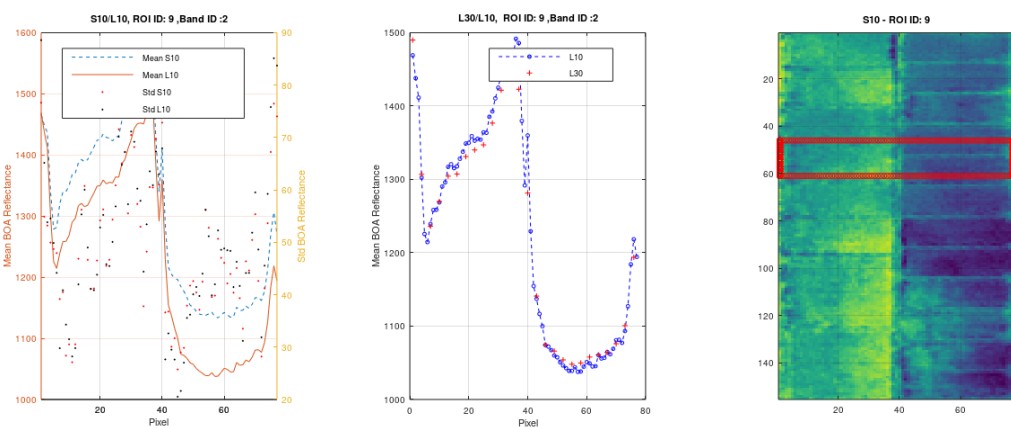

**Figure A3.** *Cont.*

B03

B04

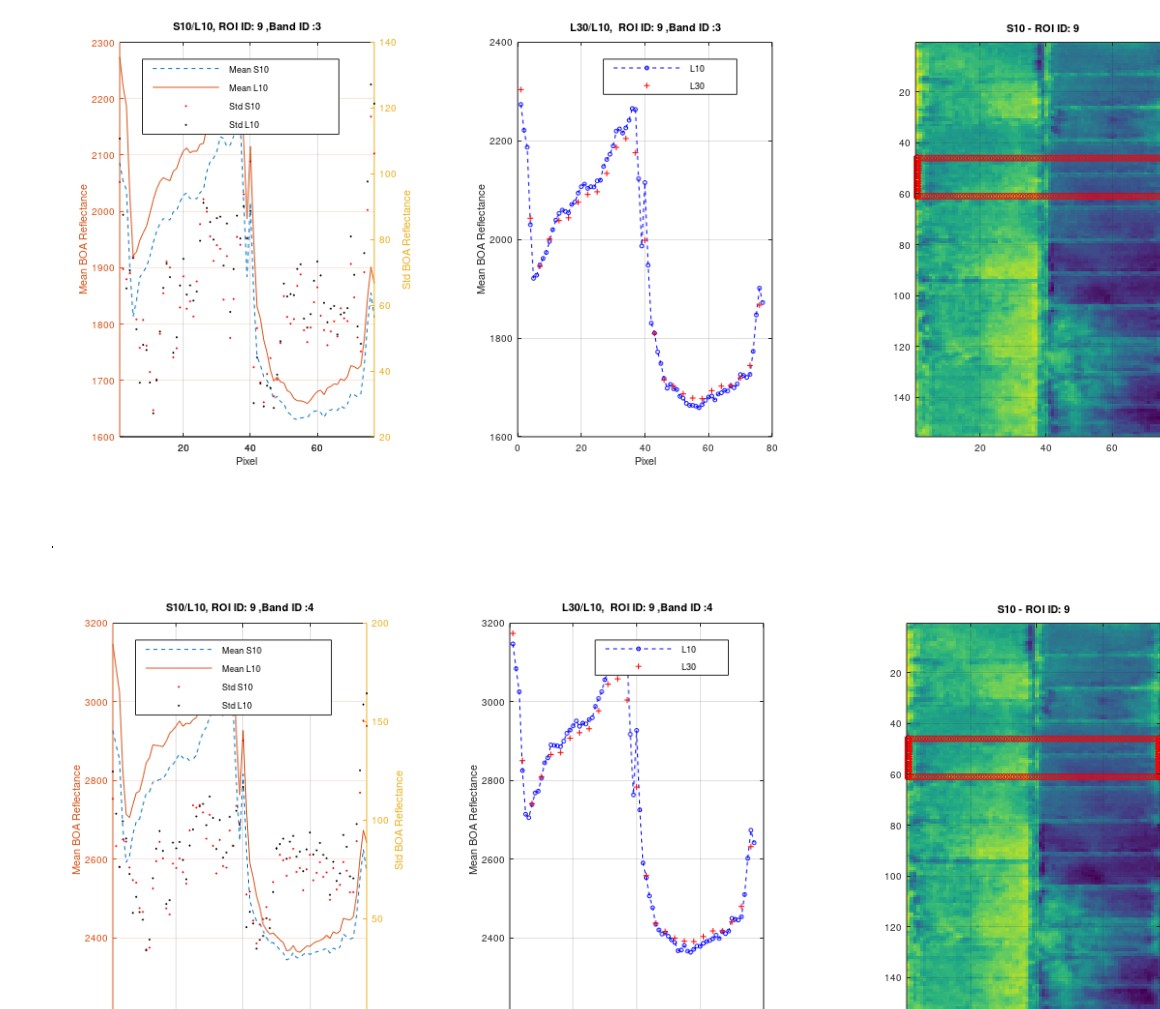

**Figure A3.** B02, B03, B04 images of ROI 9 (**right**) overplayed with image area (red box) used to compute mean edge profile in pixel direction (12SVB MGRS tile). For each band, the LS8 L2H/LS8 L2F plot (**middle**) confirms agreement between values, and S2 L2F/LS8 L2F (**left**) confirms that the synthetic LS8 L2F mean edge profile matches the S2 L2F mean edge profile. Greater error (<0.02 BOA) is observed for B02, which might be due to the inter-calibration between the two sensors.

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
