# Peer review of "Sen2Like: Paving the Way towards Harmonization and Fusion of Optical Data"

_remotesensing, doi:10.3390/rs14163855_

Round 1

Reviewer 1 Report

A relevant topic, well-structured manuscript and clear presentation of methods and results. 

Few minor comments are in the following

  • Figure 8: add legend for the L2A SCL map. Can you also add the S2L validity mask/map?
  • Figure 12: why use bar plots for the correlation results ?
  • Figure 16: what is in graph (a) and what in (b)? move legend to the right in 16(b).
  • Line 333: unfinished sentence - listed in Table 3 
  • There are grammar errors throughout the paper, some long sentences and / or incorrect sentence structure. 
  • Some references are wrong, e.g. in page 7 and page 18 

Author Response

Dear Reviewer 1,

First of all, we would like to thank for your valuable contributions to our manuscript. We have modified the paper according to your suggestions as well as the inputs of other reviewers. Please see our responses to your comments below. We would be grateful if you could review the modified version.

Kind regards,

Authors

A relevant topic, well-structured manuscript and clear presentation of methods and results. 

Authors: Thank you for your valuable contributions.

Few minor comments are in the following

  • Figure 8: add legend for the L2A SCL map. Can you also add the S2L validity mask/map?

Authors: Figures have been reprocess adding S2L validity mask and L2A SCL Legend. “Figure 8: Bottom Left: Sen2like Validity Mask, Bottom Right: L2A SCL Legend”.

  • Figure 12: why use bar plots for the correlation results ?

Authors: We could as well use a table, but considered a Figure as more readable.

  • Figure 16: what is in graph (a) and what in (b)? move legend to the right in 16(b).

Authors: We added the explanation to the Figure caption as “(a) overall view (b) zoomed view over VZA between 0° - 45°

  • Line 333: unfinished sentence - listed in Table 3

Authors: Corrected. Thank you.

  • There are grammar errors throughout the paper, some long sentences and / or incorrect sentence structure. 

Authors: We have reviewed the paper for English and incorporated the necessary revisions. Thank you.

  • Some references are wrong, e.g. in page 7 and page 18 

Authors: Corrected. Thank you.

Reviewer 2 Report

This paper presents the sequence of methods implemented to harmonize and fuse Sentinel S2 products with Landsat L8/L9. The methods are not new, but the level of processing phases to achieve the target products is high. In general, the manuscript is well written and the methods are well described, though some parts are difficult to follow, mainly in the results part. In this sense, some questions could be better explained or described. 

More specifically:

A. The first concern is about the BRDF correction.

In line 332, something is missing.

It seems that lines 379-384 and 385-392 are redundant. Please, check this extent.

B. Regarding the validation results for the fusion approach (lines 581-583), a better explanation of the “Accuracy Precision Uncertainty (APU) statistical method’ should be provided, distinguishing between these three quality metrics. It would be worth giving some useful references.    

Author Response

Dear Reviewer 2,

First of all, we would like to thank for your valuable contributions to our manuscript. We have modified the paper according to your suggestions as well as the inputs of other reviewers. Please see our responses to your comments below. We would be grateful if you could review the modified version.

Kind regards,

Authors

This paper presents the sequence of methods implemented to harmonize and fuse Sentinel S2 products with Landsat L8/L9. The methods are not new, but the level of processing phases to achieve the target products is high. In general, the manuscript is well written and the methods are well described, though some parts are difficult to follow, mainly in the results part. In this sense, some questions could be better explained or described. 

 Authors: Thank you for your valuable contributions.

More specifically:

  1. The first concern is about the BRDF correction.

In line 332, something is missing.

Authors: Corrected. Thank you.

It seems that lines 379-384 and 385-392 are redundant. Please, check this extent.

Authors: Thank you. Corrected.

  1. Regarding the validation results for the fusion approach (lines 581-583), a better explanation of the “Accuracy Precision Uncertainty (APU) statistical method’ should be provided, distinguishing between these three quality metrics. It would be worth giving some useful references.    

Authors: Thank you. We added a reference for this purpose.
“Vermote, E.F., Kotchenova, S.Y., 2008. Atmospheric correction for the monitoring of land surfaces. J. Geophys. Res. 113, D23S90. https://doi.org/10.1029/2007JD009662 .”
And update explanations.

Reviewer 3 Report

This paper presents the Sen2Like (S2L) implementation framework, an open source tool developed for data harmonization and fusion to produce Analysis Ready Data (ARD).

In section 2 when discussing data fusion, reference is made to a decomposition between contour and plot where the S2L approach bases the decomposition.
It is recommended to better define what method and algorithms are used for contour and plot classification in large- and small-scale scenes.

Better format the differences between bands B02, B03, B04, etc. in Table 4.

Add a table in Section 4 highlighting which multimission datasets the S2L framework can manage.

Reference is made in the conclusions of a potential improvement of model training processes in artificial intelligence.
Provide an example or remove this topic from the conclusions.

Author Response

Dear Reviewer 3,

First of all, we would like to thank for your valuable contributions to our manuscript. We have modified the paper according to your suggestions as well as the inputs of other reviewers. Please see our responses to your comments below. We would be grateful if you could review the modified version.

Kind regards,

Authors

This paper presents the Sen2Like (S2L) implementation framework, an open source tool developed for data harmonization and fusion to produce Analysis Ready Data (ARD).

Authors: Thank you for your valuable contributions.

In section 2 when discussing data fusion, reference is made to a decomposition between contour and plot where the S2L approach bases the decomposition.
It is recommended to better define what method and algorithms are used for contour and plot classification in large- and small-scale scenes.

Authors: Thank you for your comment. The sentences referring to the contour and texture classification provide rather a generic explanation of the fact (Lines 468-473) and the assumptions made for the data fusion process. Thus, no further object classification is carried out.

Better format the differences between bands B02, B03, B04, etc. in Table 4.

Authors: Thank you. We formatted the Table as proposed.

Add a table in Section 4 highlighting which multimission datasets the S2L framework can manage.

Authors: Thank you. Currently the tool can utilize Sentinel-2 and Landsat as explained in the manuscript (“The processing workflows contain the following configurable steps depending on the input product type (i.e., S2 L1C, S2 L2A, S2 L2A MAIA, LS8/LS9 Collection 1/Collection 2”). The datasets depend on the product levels and the availability of the calibration data. Therefore, adaptation of other missions is future work and the provision of a Table on potential missions might be confusing for the reader.

Reference is made in the conclusions of a potential improvement of model training processes in artificial intelligence. Provide an example or remove this topic from the conclusions.

Authors: Thank you. We added an example and a reference.
